# Prediction of hypertension using traditional regression and machine learning models: A systematic review and meta-analysis

**Mohammad Ziaul Islam Chowdhury**[1,2,3], **Iffat Naeem**[1], **Hude Quan**[1], **Alexander A. Leung**[1,4], **Khokan C. Sikdar**[5], **Maeve O'Beirne**[2], **Tanvir C. Turin**[1,2]*

1 Department of Community Health Sciences, Cumming School of Medicine, University of Calgary, Calgary, Alberta, Canada, 2 Department of Family Medicine, Cumming School of Medicine, University of Calgary, Calgary, Alberta, Canada, 3 Department of Psychiatry, Cumming School of Medicine, University of Calgary, Calgary, Alberta, Canada, 4 Department of Medicine, Cumming School of Medicine, University of Calgary, Calgary, Alberta, Canada, 5 Health Status Assessment, Surveillance, and Reporting, Public Health Surveillance and Infrastructure, Population, Public and Indigenous Health, Alberta Health Services, Calgary, Alberta, Canada

* chowdhut@ucalgary.ca

**Data Availability Statement:** All relevant data are within the paper and its Supporting Information files.

## Abstract

### Objective

We aimed to identify existing hypertension risk prediction models developed using traditional regression-based or machine learning approaches and compare their predictive performance.

### Methods

We systematically searched MEDLINE, EMBASE, Web of Science, Scopus, and the grey literature for studies predicting the risk of hypertension among the general adult population. Summary statistics from the individual studies were the C-statistic, and a random-effects meta-analysis was used to obtain pooled estimates. The predictive performance of pooled estimates was compared between traditional regression-based models and machine learning-based models. The potential sources of heterogeneity were assessed using meta-regression, and study quality was assessed using the PROBAST (Prediction model Risk Of Bias ASsessment Tool) checklist.

### Results

Of 14,778 articles, 52 articles were selected for systematic review and 32 for meta-analysis. The overall pooled C-statistics was 0.75 [0.73–0.77] for the traditional regression-based models and 0.76 [0.72–0.79] for the machine learning-based models. High heterogeneity in C-statistic was observed. The age (p = 0.011), and sex (p = 0.044) of the participants and the number of risk factors considered in the model (p = 0.001) were identified as a source of heterogeneity in traditional regression-based models.

**Funding:** The authors received no specific funding for this work.

**Competing interests:** The authors have declared that no competing interests exist.

**Abbreviations:** PROBAST, Prediction model Risk Of Bias ASsessment Tool; MeSH, Medical Subject Headings; HKSJ, Hartung-Knapp-Sidik-Jonkman; CI, Confidence interval; SE, Standard error; AUC, Area under the receiver operating characteristic curve; BMI, Body mass index; DBP, Diastolic blood pressure; SBP, Systolic blood pressure; ROB, Risk of bias; ANN, Artificial neural network; FHRS, Framingham hypertension risk model; O/E ratio, Observed/Expected ratio; RCT, Randomized controlled trial.

## Conclusion

We attempted to provide a comprehensive evaluation of hypertension risk prediction models. Many models with acceptable-to-good predictive performance were identified. Only a few models were externally validated, and the risk of bias and applicability was a concern in many studies. Overall discrimination was similar between models derived from traditional regression analysis and machine learning methods. More external validation and impact studies to implement the hypertension risk prediction model in clinical practice are required.

## Introduction

Hypertension is a common medical condition affecting about 1 in 4 people [1] and is a significant risk factor for heart attack, stroke, kidney disease, and mortality [2]. Hypertension has been linked to 13% of deaths globally [3] and is a significant health burden that affects all population segments. Considering the high prevalence and global burden, hypertension prevention, and control strategies need to be a top priority. Hypertension can be prevented by applying strategies that target the general population or individuals and groups at higher risk for hypertension [4]. The need for early identification of at-risk individuals who could benefit from preventive interventions has led to a growing interest in hypertension risk prediction.

Predicting the risk of developing hypertension through modeling can help identify important risk factors contributing to hypertension, provide reasonable estimates about future hypertension risk [5], and help identify high-risk individuals targeted for healthy behavioral changes and medical treatment to prevent hypertension [6–8]. Many prediction models have been developed to predict the risk of hypertension in the general population over the years. Models were developed using either a traditional regression-based approach or a modern machine learning approach. Although machine learning approaches are known to produce better predictive performance, their performance often varies, and it is not clear if they perform better than the traditional regression-based models in predicting hypertension. Through a systematic review and subsequent meta-analysis, a pooled synthesis of performance measures of different models produced in multiple studies can be compared and measured [9]. This methodology provides an overview of these models' predictive ability and allows the models' performance measures based on the reported data to be explored quantitatively [9]. Two prior studies systematically analyzed hypertension risk prediction models in adults [10, 11]. Both studies performed a narrative synthesis of the evidence to summarize hypertension prediction models' existing knowledge, and one study also performed a meta-analysis without assessing heterogeneity. None of the prior studies stratified models according to how they were developed. This stratification is important because there are inherent differences in these two types of models' developmental methods in computation, complexity, interpretability, and accuracy. A formal assessment of study quality was also absent in prior studies. In addition to these two prior reviews, a systematic review was also carried out on prediction models to classify children at an elevated risk of developing hypertension [12].

With this in mind, we aimed to 1) systematically review the literature to identify hypertension risk prediction models that have been applied to the general adult population and the risk factors that were considered in those models; 2) characterize the study populations in which these models were derived and validated, 3) compare the predictive performance of traditionally developed regression-based models and machine learning models, and 4) assess the quality of these prediction models to better inform the selection of models for clinical implementation.

## Materials and methods

### Data sources and searches

We conducted a systematic review and meta-analysis to identify existing hypertension risk prediction models and associated risk factors and evaluated the models' predictive performance. We searched MEDLINE, EMBASE, Web of Science, and Scopus (each from inception to December 2020) to identify studies predicting the risk of incident hypertension in the general adult population. Google Scholar and ProQuest (theses and dissertations) were searched for grey literature. Additionally, we explored the reference lists of all relevant articles. The search strategy focused on two key concepts: hypertension and risk prediction. We used proper free-text words and Medical Subject Headings (MeSH) terms to identify relevant studies for each key concept. Certain text words were truncated, or wildcards were used when required. The Boolean operators "AND", "OR", and "NOT" were used to combine the words and MeSH terms. A detailed search strategy for MEDLINE is provided in S1 Table.

### Eligibility criteria

Although risk prediction models are generally developed using a cohort-based study design with follow-up information, we considered all types of study designs, anticipating that machine learning-based models may use other types of study design. Only original studies were included in this review: this excluded reviews, editorials, commentaries, and letters to the editor. Studies written in languages other than English and French were also excluded. The Population, Prognostic Factors (or models of interest), and Outcome [13] framework was used to outline eligibility criteria.

**Population.** The study population consisted of people free of hypertension at baseline and those around which hypertension risk prediction models were developed. No restrictions were imposed on the geographic region, time, or gender of the study participants. Nevertheless, only models developed on the adult population were considered, as outcome essential hypertension is expected in adults.

**Prognostic factors (or models of interest).** We considered studies where risk prediction models for hypertension in the general adult population were developed. Studies that focused solely on the added predictive value of new risk factors to an existing prediction model, studies presenting a prediction model developed in patients with previous hypertension, or studies that derived risk prediction tools other than score-type tools (e.g., risk charts) were not considered. Further, we did not consider studies that only assessed bivariate association between predictors and hypertension. Instead, we focused on those studies where risk prediction models for hypertension were built incorporating risk factors that demonstrated significant prognostic contribution in predicting incident hypertension. When a model was assessed on more than one external population, information from all reported models was considered. However, when the model was presented both in a derivation and validation cohort, only data from the validation cohort were considered for meta-analysis.

**Outcome.** Our outcome of interest was hypertension, and we considered all definitions of hypertension to capture the maximum number of studies.

### Study selection

Two reviewers (MC and IN) independently identified eligible articles using a two-step process. First, the title and abstracts of non-duplicated records were screened by two reviewers. Studies retained (based on eligibility criteria) during this stage of screening went to a full-text screening. Full-text articles were further screened for eligibility by the same two reviewers

independently. Lastly, articles containing extractable data on hypertension prediction models and hypertension risk factors were selected for data extraction. Inter-rater reliability (Kappa coefficient) was estimated to measure agreement between the independent reviewers. Any disagreement between reviewers was resolved through consensus.

## Data extraction

Two reviewers (MC and IN) independently extracted data from each study using standardized forms. We classified the identified models into two categories: models developed using a traditional regression-based approach and models developed using machine learning algorithms. Separate data extraction sheets were used for each model type and included study name, the location where the model was developed/location of data used for the model developed and participants' ethnicity, study design used, sample size, age, and gender of the study participants, risk factors included in the model, number of events and total participants, an outcome considered, the definition used for hypertension, duration of follow-up, modeling method used, measures of discrimination and calibration of the prediction model, and the validation of the prediction model. In a separate form, information about the externally validated hypertension risk prediction models was extracted, including study name/model validated, the total number of validation studies, location of the validation study, follow-up period, number of events, and total participants, the definition of outcome and discrimination and calibration of the model. We also extracted information about risk factors, particularly how many times a specific risk factor was considered in the models. Each reviewer assessed study quality according to the Prediction model Risk Of Bias ASsessment Tool (PROBAST) checklist [14, 15]. The PROBAST is designed to evaluate the risk of bias and concerns regarding diagnostic and prognostic prediction model studies' applicability. The PROBAST contains 20 questions under four domains: participants, predictors, outcome, and analysis, facilitating judgment of risk of bias and applicability. The overall risk of bias of the prediction models was judged as "low", "high", or "unclear," and overall applicability of the prediction models was considered as "low concern", "high concern", and "unclear" according to the PROBAST checklist [14, 15].

## Data analysis

We summarized the number of studies identified and those included and excluded (with the reason for exclusion) from the systematic review and subsequent meta-analysis using the PRISMA flow diagram [16]. In data synthesis, we performed a meta-analysis on the performance measure of the traditional regression type's prediction modeling (e.g., logistic regression model and Cox proportional hazard regression model) and a more complicated modeling strategy (e.g., machine learning tools). Discrimination and calibration are the two most common statistical measures of predictive performance. Discrimination is commonly quantified by the concordance (C) statistic. In this review, we performed a meta-analysis on the C-statistic or AUC (area under the receiver operating characteristic curve) to evaluate the models' predictive performance and provided a comprehensive summary of the models' predictive ability. We did not undertake a meta-analysis of the calibration due to the unavailability of relevant data.

We logit transformed the C-statistics before pooling as per recommendation [17, 18] and then back-transformed the results to the original scale for interpretation. We used a random-effects meta-analysis with REML estimation and Hartung-Knapp-Sidik-Jonkman (HKSJ) confidence interval (CI) to obtain the pooled weighted average of the logit C-statistic [19]. Forest plots were generated to show the pooled C-statistic together with the 95% CI, 95% approximate prediction interval (indicates an expected performance range of the considered models

in a new population) for the summary C-statistic, the author's name, publication year, and study weights. In studies that only provided a C-statistic but no measure of its variance or confidence intervals, the standard error (SE) and 95% CI of the logit C-statistic (or area under the receiver operating characteristic curve (AUC)) was calculated using the appropriate formula [19]. However, when the C-statistics' confidence intervals (CIs) were available, standard errors (SE's) of the logit C-statistics were derived from the CIs [19]. The presence of heterogeneity (primarily due to differences in the study setting, participants, and methodology) was assessed using Cochran's Q statistic and quantified with the $I^2$ statistic. A p-value of less than 0.05 was considered statistically significant heterogeneity and was categorized as low, moderate, and high when the $I^2$ values were below 25%, between 25% and 75%, and above 75%, respectively [20]. Sources of heterogeneity were further explored using meta-regression and stratified analyses according to modeling type and study characteristics (sex of the participants, age of the participants, number of risk factors considered in the model, sample size considered in the model, and ethnicity of the study participants). We calculated 95% prediction intervals to provide a likely range of performance of a prediction model in a new population and setting. We did not assess publication bias by any statistical tests or funnel plot asymmetry. We used Stata version 16.1 (StataCorp LP, College Station, TX, USA) to perform statistical analysis using the following commands: meta, metan and metareg.

## Results

### Study identification and selection

We identified 14,730 articles through our electronic database search and an additional 48 articles through our grey literature search. After removing duplicates, titles, and abstracts screening and full-text screening 52 articles were finally selected for the systematic review. Within the chosen final studies, 32 studies provided sufficient information for synthesis through a meta-analysis. The detailed study selection process is summarized in Fig 1. Agreement between reviewers on the initial screening and final articles eligible for inclusion in the systematic review was good (κ = 0.81, and κ = 0.89, respectively). A total of 117 models were identified from the finally selected articles predicting the risk of hypertension in the general adult population, of which 75 were developed using traditional regression-based modeling and 42 using machine learning tools.

### Study characteristics of traditional regression-based models

Study characteristics of traditional regression-based models are presented in Tables 1 and 2. A total of 573,268 participants were used to develop 75 traditional models in 34 studies. Models mainly were developed either in white Caucasian or Asian populations. There was no model derived from African populations and only one [21] from Latin American populations. Two studies considered only male participants, one study considered only female participants, and the remaining studies considered both to develop the models. The number of risk factors considered to create the models ranged from 1 to 19, with a median of 7 risk factors per model. Age was the most common risk factor considered in 61 models, followed by body mass index (BMI) (32 models), diastolic blood pressure (DBP) (28 models), systolic blood pressure (SBP) (27 models), and sex (21 models). The distribution of the conventional risk factors considered in the different models is presented in Fig 2A. Duration of follow-up time (mean/median/total) considered to develop the models varied between 1.6 years to 30 years. The age of the study participants ranged from 15 to 90 years. SBP ≥ 140 mm Hg, DBP ≥ 90 mm Hg, or use of antihypertensive medication was the standard definition used to define hypertension in almost all the studies, except one study where SBP ≥ 130 mm Hg, DBP ≥ 80 mm Hg, or use of

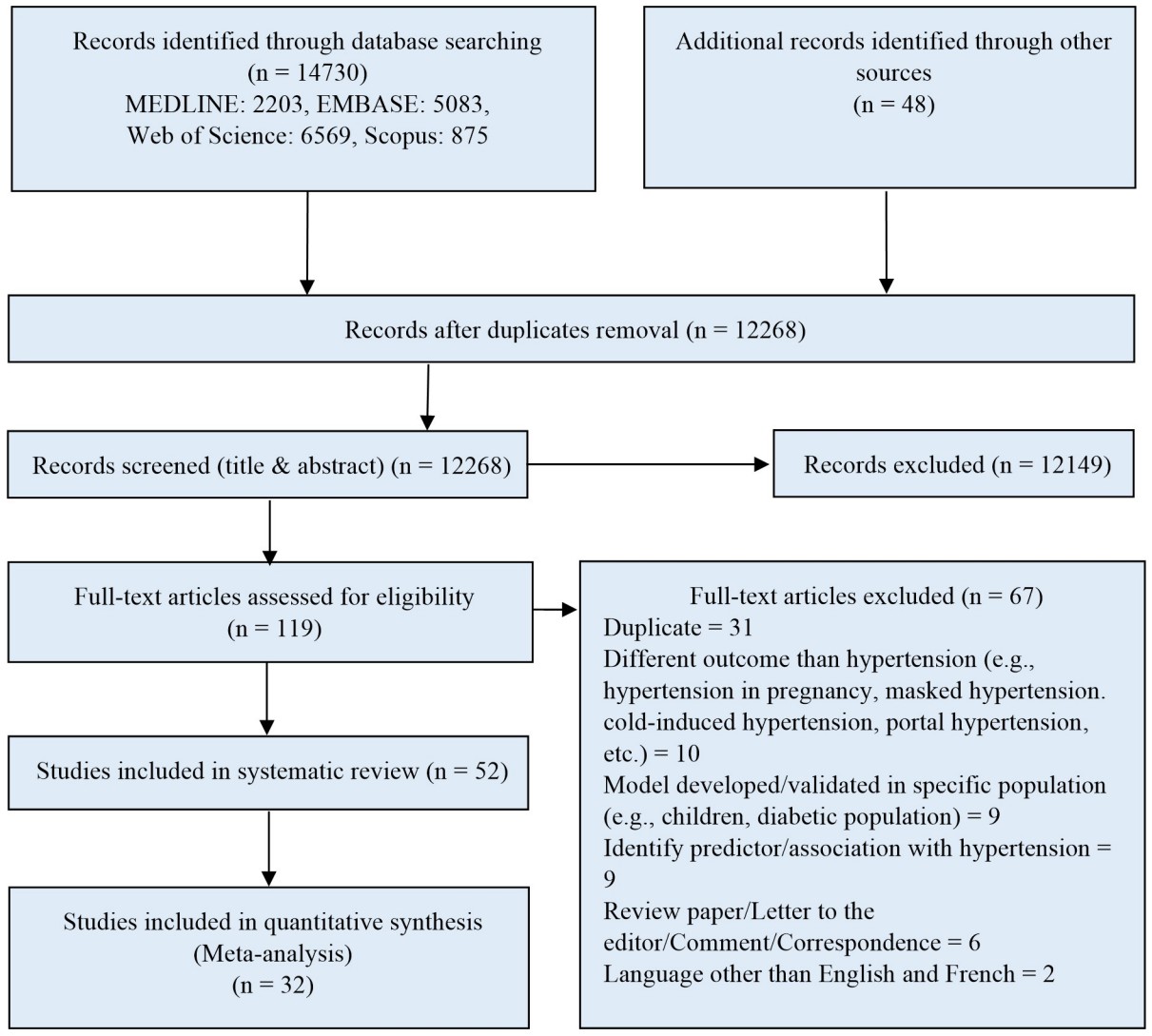

**Fig 1. PRISMA diagram for systematic review of studies presenting hypertension prediction models developed in the general population.**

any antihypertensive drug was used. Logistic regression was the most used methodology to develop the model (15 studies), followed by Cox proportional-hazards regression (11 studies) and Weibull regression (6 studies). Calibration of the prediction model was not reported by most of the studies (19 studies). Studies those reported calibration measures (15 studies) were mainly using the Hosmer-Lemeshow test. Discrimination was assessed using the C-statistic (or AUC) and reported by almost all studies with values ranging from 0.57 to 0.97. Only one model was externally validated by the same study when they developed the model. Only eight models [22–29] were converted into a risk score after model development.

## Meta-analysis of traditional regression-based models

The overall pooled C-statistics of the traditional regression-based models was 0.75 [0.73–0.77] with high heterogeneity in the discriminative performance of these models ($I^2$ = 99.3, Cochran Q-statistic p < 0.001) (Fig 3). Stratified pooled results by modeling type showed pooled C-

**Table 1. Characteristics of included studies that describe traditional regression-based hypertension prediction models.**

| Study | Location Model Developed/ Ethnicity | Study Design | Age | Gender | Events (n)/Total Participants (N) | Definition of Outcome Predicted/ Hypertension | Duration of Follow-up |
|---|---|---|---|---|---|---|---|
| Pearson et al. [41] 1990 | USA/Mixed, mainly Whites | Prospective cohort | ≤ 25 years | Male only | 114/1130 | Self-reported use of blood pressure-lowering medications | 30 years |
| Parikh et al. [22] 2008 | USA/Mainly Whites | Prospective cohort | 20–69 years | Both | 796/1717 | SBP ≥ 140 mmHg or DBP ≥ 90 mmHg or use of BP-lowering medications | Median 3.8 years |
| Paynter et al. [42] 2009 | USA/ Whites and Blacks | Prospective cohort | 45–64 years | Female only | Derivation cohort: 1935/9427 Validation cohort: 1068/5395 | Self-report or SBP ≥ 140 mmHg or DBP ≥ 90 mmHg | 8 years |
| Kivimäki et al. [43] 2009 | England/Mainly Whites | Prospective cohort | 35–68 years | Both | 1258/8207 | SBP ≥ 140 mmHg or DBP ≥ 90 mmHg or use of BP-lowering medications | Median 5.6 years |
| Kivimäki et al. [44] 2010 | England/Mainly Whites | Prospective cohort | 36–68 years | Both | Derivation cohort: 614/4135 Validation cohort: 438/2785 | SBP ≥ 140 mmHg or DBP ≥ 90 mmHg or use of antihypertensive medications | Median 5.8 years |
| Kshirsagar et al. [45] 2010 | USA/Mixed but mainly Whites | Prospective cohort | 45–64 years | Both | 3795/11,407 (7610 for derivation sample and 3692 for the validation sample) | SBP ≥ 140 mmHg or DBP ≥ 90 mmHg or reported use of BP-lowering medications | Up to 9 years |
| Bozorgmanesh et al., [25] 2011 | Iran/Asians | Prospective cohort | ≥ 20 years | Both | 805/4656 | SBP ≥ 140 mmHg or DBP ≥ 90 mmHg or reported use of BP-lowering medications | 6 years |
| Chien et al. [24] 2011 | Taiwan/Chinese | Prospective cohort | ≥ 35 years | Both | 1029/2506 | SBP ≥ 140 mmHg or DBP ≥ 90 mmHg or reported use of BP-lowering medications | Median 6.15 years |
| Fava et al. [46] 2013 | Sweden/Whites | Prospective cohort | Middle-aged | Both | NR/10,781 | SBP ≥ 140 mmHg or DBP ≥ 90 mmHg or reported use of BP-lowering medications | Over average 23-years |
| Lim et al. [30] 2013 | Korea/Asians | Prospective cohort | 40–69 years | Both | 819/4747. Derivation cohort: 483/2840 Validation cohort: 336/1907 | SBP ≥ 140 mmHg or DBP ≥ 90 mmHg or reported use of BP lowering medications | 4 years |
| Choi et al. [47] 2014 | USA/Mexicans | Prospective cohort | NR | Both | NR/443 | SBP >140 mm Hg, DBP >90 mm Hg, or use of antihypertensive medication | NR |
| Lim et al. [48] 2015 | Korean/Asians | Prospective cohort | 40–69 years | Both | NR/5632 | SBP ≥140 mm Hg or DBP ≥90 mm Hg or use of antihypertensive medication | 4-year |
| Otsuka et al. [23] 2015 | Japan/Asians | Prospective cohort | 19–63 years | Male only | 1633/15,025 | SBP ≥140 mm Hg or DBP ≥90 mm Hg or use of antihypertensive medication | Median 4 years |
| Asgari et al. [49] 2015 | Iran/Asians | Prospective cohort | ≥ 20 years | Both | ISH: 235/4574 IDH: 470/4809 | ISH: SBP ≥ 140 mmHg and DBP < 90 mmHg IDH: SBP <140 mmHg and DBP ≥ 90 mmHg | ISH: Median 9.57 years, IDH: Median 9.62 years |
| Sathish et al. [29] 2016 | India/Asians | Prospective cohort | 15–64 years | Both | 70/297 | SBP ≥140 mm Hg or DBP ≥90 mm Hg or use of antihypertensive medication | Mean 7.1 years |
| Lee et al. [50] 2015 | Korea/Asians | Prospective cohort | 40–69 years | Both | Men: 384/2128 Women: 374/2326 | SBP ≥140 mm Hg or DBP ≥90 mm Hg or use of antihypertensive medication | 4 years |
| Lee et al. [51] 2014 | Korea/Asians | Cross-sectional | 21–85 years | Both | NR/12,789 | SBP ≥ 140 mmHg and/or DBP ≥ 90 mmHg or physician-diagnosed hypertension | NR |

*(Continued)*

**Table 1.** (Continued)

| Study | Location Model Developed/ Ethnicity | Study Design | Age | Gender | Events (n)/Total Participants (N) | Definition of Outcome Predicted/ Hypertension | Duration of Follow-up |
|---|---|---|---|---|---|---|---|
| Kanegae et al. [32] 2018 | Japan/Asians | Prospective cohort | 18–83 years | Both | 7402/63,495 | SBP/DBP ≥ 140/90 mm Hg and/or the initiation of antihypertensive medications with self-reported hypertension | Mean 3.4 years |
| Chen et al. [52] 2016 | China/Asians | Prospective cohort | Average age 41.73 years (men), 39.49 years (women) | Both | 2021 (men), 764 (women) 7537 (men), 4960 (women) | First occurrence at any follow-up medical check-up of SBP > 140 mm Hg or DBP > 90 mm Hg or of the person taking antihypertensive medication | Median 4.0 years |
| Díaz-Gutiérrez et al. [28] 2019 | Spain/Spanish | Prospective cohort | Age presented according to the number of healthy lifestyle factors | Both | 1406/14057 | SBP ≥ 130 mmHg, DBP≥ 80 mmHg, or use of any antihypertensive drug | Median 10.2 years |
| Wang et al. [53] 2018 | China/Asians | Longitudinal | 18–90 years | Both | 882/5265 (derivation) NR/1597 (validation) | Taking antihypertensive drugs or SBP at least 140 mmHg or DBP at least 90 mmHg | Average follow-up of 8.05 ± 5.27 years |
| Niiranen et al. [54] 2016 | Finland/Whites | Prospective cohort | ≥ 30 years | Both | NR/2045 | BP ≥ 140/90 mm Hg and/or antihypertensive medication | 11 years |
| Yeh et al. [55] 2001 | Taiwan/Chinese | Prospective cohort | ≥ 20 years | Both | 88/2374 | SBP ≥140 mm Hg or DBP ≥90 mm Hg | Average 3.23 years |
| Syllos et al. [21] 2020 | Brazil/South Americans | Prospective cohort | 35–74 years | Both | 1088/8027; Derivation: 4825 Validation: 3202 | SBP ≥ 140 mm Hg, DBP ≥ 90 mm Hg or the use of blood pressure-lowering medications | 4 years |
| Wang et al. [27] 2020 | China/Asians | Prospective cohort | ≥ 18 years | Both | 1658/9034 | SBP ≥ 140 mm Hg, DBP ≥ 90 mm Hg or the use of blood pressure-lowering medications | Median 6 years |
| Xu et al. [56] 2019 | China/Asians | Prospective cohort | 35–74 years | Both | 1036/4796 (Training) | SBP ≥ 140 mm Hg and/or DBP ≥ 90 mm Hg, and/or a diagnosis of hypertension by a physician and currently receiving anti-hypertension treatment | 6 years |
| Kadomatsu et al. [26] 2019 | Japan/Asians | Prospective cohort | Mean age 51.3 years | Both | 324/3936 | SBP ≥ 140 mm Hg, DBP ≥ 90 mm Hg, or use of antihypertensive medication | Median 5 years |
| Wang et al. [57] 2015 | USA/Multi-ethnic | Telephone-based health survey | ≥ 18 years | Both | NR/308,711 | NR | NR |
| Muntner et al. [58] 2010 | USA/ Multi-ethnic (Whites, Blacks, Hispanics, and Asians) | NR | 45–84 years | Both | 849/3013 | The first study visit, subsequent to baseline, at which SBP ≥ 140 mm Hg and/or DBP ≥ 90 mm Hg and/ or the initiation of antihypertensive medication | Median of 1.6 years and 4.8 years |
| Ture et al. [59] 2005 | Turkey/ Europeans | Retrospective | Average 48.2 years (hypertension) 46.5 (control) | Both | 694 (452 patients with hypertension and 242 controls) | Average of 3 or more DBP measurements on at least 3 subsequent visits is ≥ 90 mmHg or when the average of multiple SBP readings on 3 or more subsequent visits is consistently ≥ 140 mmHg | NR |
| Yamakado et al. [60] 2015 | Japan/Asians | Prospective cohort | ≥ 20 years | Both | 424/2637 | SBP ≥ 140 mm Hg or DBP ≥ 90 mm Hg or use of antihypertensive medication | 4 years |
| Qi et al. [61] 2014 | China/Asians | Case-control | Case: 64.48 ± 8.53 years; Control: 64.23 ± 10.13 years | Both | Patients: NR/1009 Controls = NR/756 | SBP ≥ 140 mm Hg or DBP ≥ 90 mm Hg or use of antihypertensive medication | NR |

(*Continued*)

**Table 1.** (Continued)

| Study | Location Model Developed/ Ethnicity | Study Design | Age | Gender | Events (n)/Total Participants (N) | Definition of Outcome Predicted/ Hypertension | Duration of Follow-up |
|---|---|---|---|---|---|---|---|
| Lu et al. [62] 2015 | China/Asians | Prospective cohort | 35–74 years | Both | 2559/7724 | SBP ≥ 140 mm Hg or DBP ≥ 90 mm Hg or use of antihypertensive medication | Mean 7.9 years |
| Zhang et al. [63] 2015 | China/Asians | Prospective cohort | 18–88 years | Both | 3793/17,471 | SBP ≥ 140 mm Hg or DBP ≥ 90 mm Hg or use of antihypertensive medication | 5 years |

NR, not reported; SBP, systolic blood pressure; DBP, diastolic blood pressure; BP, blood pressure; ISH, isolated systolic hypertension; IDH, isolated diastolic hypertension

statistics were 0.73 [0.69–0.77], 0.77 [0.74–0.81], 0.73 [0.69–0.78], and 0.77 [0.75–0.79] for Cox, logistic, repeated Poisson, and Weibull respectively (Fig 3). The heterogeneity was still observed to be high within the different types of models (Fig 3). The 95% approximate prediction interval for the overall C-statistics was from 0.63 to 0.84.

To explore possible sources of heterogeneity in the overall pooled C-statistics, we performed a meta-regression. We initially considered the following potential sources of heterogeneity: the definition of hypertension used (the cut-off level used to define hypertension), sex of the participants in included studies (categorized as female-only, male-only, and both male and female), age of the participants (study participants below average age versus above average age), number of risk factors considered in the model (below median versus above median), sample size considered in the model (below median versus above median), and ethnicity of the study participants (Whites versus Asians). However, we excluded the definition of hypertension as a heterogeneity source, as all studies except one used the same definition for hypertension. Meta-regression identified the participants' sex, that is, being male compared to female (p = 0.044), participants' age (p = 0.011), and the number of risk factors considered in the model (p = 0.001) as potential sources of high heterogeneity in the C-statistic. Sex of the participants' when both male and female compared to female-only (p = 0.351), sample size considered in the model (p = 0.395), and ethnicity of the study participants (p = 0.899) were not identified as a statistically significant source of observed heterogeneity in the C-statistic of these models.

### Critical appraisal of traditional regression-based models

We assessed study quality using the PROBAST checklist. A detailed assessment of the risk of bias (ROB) and applicability is presented in S2 Table and Fig 4. Overall, ROB was "low" in 19 studies, "high" in 5 studies, and "unclear" in 10 studies. Overall applicability was "low concern" in 12 studies, "high concern" in 21 studies, and "unclear concern" in 1 study. Within the ROB domains, the "low" risk of bias was observed in most of the domains except the "analysis" domain, where a large portion of studies (more than 30%) was "unclear" (Fig 4). Similarly, within the applicability domains, the "participants" domain seems to be a concern, as a large portion of studies (more than 30%) were at "high concern" or "unclear concern" (Fig 4). We also presented the different PROBAST signaling questions' distribution of responses by the various studies in S1 and S2 Figs.

### Study characteristics of machine learning-based models

Study characteristics of machine learning-based models are presented in Table 3. A total of 1,211,093 participants were used to develop 42 machine learning-based models in 20 studies.

**Table 2. The features of hypertension prediction models developed using a traditional regression-based modeling approach.**

| Study | Risk Factors Included | Modeling Method | Discrimination | Calibration | Model Validation: Internal or External |
|---|---|---|---|---|---|
| Pearson et al. [41] 1990 | Age, SBP at baseline, paternal history of hypertension, and BMI | Cox regression | NR | NR | NR |
| Parikh et al. [22] 2008 | Age, sex, SBP, DBP, BMI, parental hypertension, and cigarette smoking | Weibull regression | C-statistic = 0.788 [0.733–0.803] | HL Chi-square = 4.35 (p = 0.88) | Internal, apparent |
| Paynter et al. [42] 2009 | Inclusive Model: Age, ethnicity, BMI, total grain intake, SBP, DBP, apolipoprotein B, lipoprotein (a), and C-reactive protein. Simplified Model with Lipids: Age, BMI, SBP, DBP, ethnicity, and total to HDL- cholesterol ratio Simplified Model: Age, BMI, ethnicity, SBP, and DBP | Logistic regression | Inclusive Model: C-statistic = 0.705; Simplified Model with Lipids: C-statistic = 0.705; Simplified Model: C-statistic = 0.703 | Inclusive Model: HL Chi-square = 24.6 (p = 0.002), Simplified Model with Lipids: HL Chi-square = 20.7 (p = 0.008), Simplified Model: HL Chi-square = 12.3 (p = 0.140) | Internal, split-sample 2:1 |
| Kivimäki et al. [43] 2009 | Age, sex, SBP, DBP, BMI, parental hypertension, and cigarette smoking | Weibull regression | C-statistic = 0.804 | HL Chi-square = 14.3 (p = 0.88) | Internal, split-sample 6:4 |
| Kivimäki et al. [44] 2010 | Repeat Measure BP Model: Age, sex, BMI, parental hypertension, repeat measures of BP, and cigarette smoking Average BP Model: Age, sex, BMI, parental hypertension, average BP, and cigarette smoking | Weibull regression | Repeat Measure BP Model: C-statistic = 0.799; Average BP Model: C-statistic = 0.794 | Repeat Measure BP Model: HL Chi-square = 6.5; Average BP Model: NR | Internal, split-sample 6:4 |
| Kshirsagar et al. [45] 2010 | Age, level of SBP or DBP, smoking, family history of hypertension, diabetes mellitus, BMI, female sex, and lack of exercise | Logistic regression | AUC = 0.742 (3years), 0.750 (6 years), 0.791 (9 years), and 0.775 (ever) | NR | Internal, split-sample 2:1 |
| Bozorgmanesh et al., [25] 2011 | For Women: age, waist circumference, DBP, SBP, and family history of premature CVD For Men: age, DBP, SBP, and smoking | Weibull regression | C-statistic = 0.731 [0.706–0.755] for women, C-statistic = 0.741 [0.719–0.763] for men | HL Chi-square = 7.8 (p = 0.554) for women; HL Chi-square = 8.8 (p = 0.452) for men | NR |
| Chien et al. [24] 2011 | Clinical Model: Age, gender, BMI, SBP, and DBP Biochemical Model: Age, gender, BMI, SBP, DBP, white blood count, fasting glucose, uric acid | Weibull regression | Clinical Model: AUC = 0.732 [0.712–0.752] (point based), AUC = 0.737 (coefficient based); Biochemical Model: AUC = 0.735 [0.715–0.755] (point based), AUC = 0.74 (coefficient based) | Clinical Model: HL Chi-square = 8.3, p = 0.40 (point based), 10.9, p = 0.21 (coefficient based); Biochemical Model: HL Chi-square = 13.2, p = 0.11 (point based), 6.4, p = 0.60 (coefficient based) | Internal, fivefold cross-validation |
| Fava et al. [46] 2013 | Age, sex, sex times age, heart rate, obesity, diabetes, hypertriglyceridemia, prehypertension, family history of hypertension, sedentary in spare time, problematic alcohol behavior, married or living as a couple, high-level non-manual work, smoking | Logistic regression | AUC = 0.662 [0.651–0.672] | NR | NR |
| Lim et al. [30] 2013 | Age, sex, smoking, SBP, DBP, parental hypertension, BMI | Weibull regression | AROC = 0.791 [0.766–0.817] | HL Chi-square = 4.17 (p = 0.8415) | Internal, split-sample 6:4 |

(*Continued*)

**Table 2.** (Continued)

| Study | Risk Factors Included | Modeling Method | Discrimination | Calibration | Model Validation: Internal or External |
|---|---|---|---|---|---|
| Choi et al. [47] 2014 | Age, gender, smoke, age x gender, Rs10510257 (AA), Rs10510257 (AG), Rs1047115 (GT) | GEE for marginal model and logistic random effect model for conditional model | Marginal model: AUC = 0.839 (with SNPs), 0.826 (without SNPs) Conditional model: AUC = 0.973 (with SNPs), 0.973 (without SNPs) | NR | NR |
| Lim et al. [48] 2015 | Traditional variables: age, gender, SBP, current smoking status, family history of hypertension, BMI, and one genetic variable (cGRS or wGRS derived from the 4 SNPs): rs995322, rs17249754, rs1378942, rs12945290 | Logistic regression | Derivation cohort: C-statistic = 0.810 [0.796–0.824] (model without wGRS, C-statistic = 0.811 [0.797–0.825]) (model with wGRS); Validation cohort: Mean C-statistic = 0.811 [0.809–0.816] | HL Chi-square = 6.916 (model without wGRS), HL Chi-square = 5.711 (model with wGRS) | Internal validation, fivefold cross-validation |
| Otsuka et al. [23] 2015 | Age, BMI, SBP and DBP, current smoking status, excessive alcohol intake, parental history of hypertension | Cox regression | Validation cohort: C-statistic = 0.861 [0.844–0.877] (model), C-statistic = 0.858 [0.840–0.876] (score) | Validation cohort: HL Chi-square = 15.2 (p = 0.085) (model), HL Chi-square = 9.30 (p = 0.41) (score) | Internal validation, split sample 4:1 |
| Asgari et al. [49] 2016 | ISH: Age, SBP, BMI, 2 hours post-challenge plasma glucose IDH: Age, DBP, waist circumference, marital status, gender, HDL-C | Cox regression | ISH: C-statistic = 0.91, IDH: C-statistic = 0.76 | NR | NR |
| Sathish et al. [29] 2016 | Age, sex, years of schooling, daily intake of fruits or vegetables, current smoking, alcohol use, BP, prehypertension, central obesity, history of high blood glucose | Logistic regression | AUC = 0.802 [0.748–0.856] | Hosmer-Lemeshow p = 0.940 | NR |
| Lee et al. [50] 2015 | BMI, waist circumference, waist-to-hip ratio, waist-to-height ratio | Cox regression | Men: AROC = 0.58 [0.56–0.60] (BMI), 0.62 [0.60–0.64] (WC, WHR, WHtR) Women: AROC = 0.57 [0.55–0.59] (BMI), 0.66 [0.64–0.68] (WC), 0.68 [0.66–0.70] (WHR, WHtR) | NR | NR |
| Lee et al. [51] 2014 | Women: Height, age, neckC, axillaryC, ribC, waistC, pelvicC, rib_hip, waist_hip, pelvic_hip, rib_pelvic, axillary_rib, chest_rib, axillary_chest, forehead_neck (CFS), height, weight, BMI, age, chestC, forehead_hip, waist_hip, chest_pelvic, waist_pelvic, axillary_waist, forehead_rib, neck_axillary (LR-wrapper) Men: Age, foreheadC, neckC, axillaryC, chestC, ribC, waistC, pelvicC, hipC, rib_hip, waist_hip, rib_pelvic, waist_pelvic, chest_waist, forehead_rib, chest_rib, axillary_chest, forehead_neck (CFS), height, foreheadC, neckC, axillaryC, ribC, pelvicC, forehead_hip, chest_hip, rib_hip, pelvic_hip, forehead_waist, axillary_waist, rib_waist, neck_rib, axillary_rib, chest_rib, forehead_axillary, forehead_neck, WHtR (LR-wrapper) | Logistic regression | Women: AUC = 0.713 (LR-CFS), 0.721 (LR-wrapper) Men: AUC = 0.637 (LR-CFS), 0.652 (LR-wrapper) | NR | Internal, 10-fold cross-validation |

(*Continued*)

**Table 2.** (*Continued*)

| Study | Risk Factors Included | Modeling Method | Discrimination | Calibration | Model Validation: Internal or External |
|---|---|---|---|---|---|
| Kanegae et al. [32] 2018 | Age, sex, BMI, SBP, DBP, low-density lipoprotein cholesterol, uric acid, proteinuria, current smoking, alcohol intake, eating rate, DBP by age, and BMI by age | Cox regression | C-statistic = 0.885 [0.865–0.903] | Greenwood-Nam-D'Agostino χ2 statistic = 13.6) | External validation |
| Chen et al. [52] 2016 | Men: Age, BMI, SBP, DBP, gamma-glutamyl transferase, fasting blood glucose, drinking, age x BMI, age x DBP Women: Age, BMI, SBP, DBP, fasting blood glucose, total cholesterol, neutrophil granulocyte, drinking, smoking | Cox regression | Derivation: AUC = 0.761 [0.752–0.771] (men), 0.753 [0.741–0.765] (women) Validation: AUC = 0.760 [0.751–0.770] (men), 0.749 [0.737–0.761] (women) | NR | Internal, 10-fold cross-validation |
| Díaz-Gutiérrez et al. [28] 2019 | No smoking, moderate-to-high physical activity, Mediterranean diet adherence, healthy BMI, moderate alcohol intake, and no binge drinking | Cox regression | NR | NR | NR |
| Wang et al. [53] 2018 | Age, sex, education, marriage, smoking, drinking, BMI, energy, carbo, fat, protein | Multistate Markov model | NR | NR | Temporal validation |
| Niiranen et al. [54] 2016 | Model 1: GRS Model 2: Model 1 + age + sex Model 3: Model 2 + smoking, diabetes, education, hyper-cholesterolemia, leisure-time exercise, and BMI | Multiple linear and logistic regression | C-index = 0.731 (Model 1) C-index = 0.733 (Model 3) | NR | NR |
| Yeh et al. [55] 2001 | Age, DM, and fibrinogen concentration (Men) Age and APTT (activated partial thromboplastin time) (Women) | Cox regression | NR | NR | NR |
| Syllos et al. [21] 2020 | Age, sex, educational level, parental history of hypertension, leisure-time physical activity, BMI, neck circumference, smoking, SBP, DBP | Logistic regression | AUC = 0.830 [0.810–0.849] | H-L Chi-square = 8.22, p = 0.41 | Internal, split sample 6:4 ratio |
| Wang et al. [27] 2020 | Age, parental hypertension, SBP, DBP, BMI, and age by BMI | Logistic regression | C-index = 0.795 [0.7733–0.810] (Training set), C-index = 0.7914 [0.773–0.809] (Testing set) | H–L Chi-square = 7.747, P = 0.459 (Training set) H–L Chi-square = 14.366, P = 0.073 (Testing set) | Internal, Bootstrap validation |
| Xu et al. [56] 2019 | M1 Model: Age, SBP, DBP, hypertension parental history, WC, interaction item of age with WC, and interaction item of age with DBP W1 Model: Age, SBP, DBP, WC, fruit and vegetable intake, hypertension parental history, interaction item of age with WC, and interaction of age with DBP were included in W1 model | Cox regression | Testing Set Men: AUC = 0.771 [0.750–0.791] (M1) Testing Set Women: AUC = 0.765 [0.746–0.783] (W1), 0.764 [0.746–0.783] (W2) | Testing Set Men: Modified Nam-D'Agostino test Chi-square = 6.305, p = 0.708 (M1) Testing Set women: Modified Nam-D'Agostino test Chi-square = 6.783, p = 0.147(W1); 7.404, p = 0.115 (W2) | Internal, 10-fold cross-validation in training data and external in the testing data |
| Kadomatsu et al. [26] 2019 | Age, sex, BMI, current smoking habit, ethanol consumption, presence of DM, parental hypertension history, SBP, DBP | Logistic regression | AUC = 0.826 [0.804–0.848] (Entire cohort validation) Median AUC = 0.83 [0.828–0.832] (Cross-validation) | H–L Chi-square = 7.06, p = 0.53, (Entire cohort validation); H–L Chi-square = 12.2 (Cross-validation) | Internal, split-sample cross-validation 6:4 ratio |

(*Continued*)

**Table 2.** (Continued)

| Study | Risk Factors Included | Modeling Method | Discrimination | Calibration | Model Validation: Internal or External |
|---|---|---|---|---|---|
| Wang et al. [57] 2015 | Exercise, diabetes, hyperlipemia, age, marriage, education, income, weight, height, sex, smoke, drink | Logistic regression | Accuracy, sensitivity, specificity, and AUC. AUC = 0.74±0.001 (logistic), Accuracy = 71.96% (logistic) | NR | Internal, split sample 7:3 ratio |
| Muntner et al. [58] 2010 | SBP-alone model (7 SBP categories) Age-specific categories of DBP model (20 categories) | Repeated-measures Poisson regression model | SBP model: C-statistic = 0.768 [0.751–0.785] (1.6 years follow-up), 0.773 [0.775–0.791] (4.8 years follow-up); Age-specific DBP Model: C-statistic = 0.699 [0.681–0.717] (1.6 years follow-up), 0.691 [0.671–0.711] (4.8 years follow-up) | NR | NR |
| Ture et al. [59] 2005 | Age, sex, family history of hypertension, smoking habits, lipoprotein (a), triglyceride, uric acid, total cholesterol, and BMI | Logistic regression, Flexible discriminant analysis, multivariate additive regression splines (degree 1), multivariate additive regression splines (degree 2) | Sensitivity, specificity, and predictive rate (PR) | NR | Internal, split sample 3:1 ratio |
| Yamakado et al. [60] 2015 | PFAA Index 1: Leucine, alanine, tyrosine, asparagine, tryptophan, and glycine; PFAA Index 2: Isoleucine, alanine, tyrosine, phenylalanine, methionine, and histidine | Logistic regression | NR | NR | Internal, LOOCV and validation in a cohort dataset |
| Qi et al. [61] 2014 | rs17030613, rs16849225, rs1173766, rs11066280, rs35444, rs880315, rs16998073, rs11191548, rs17249754 | Logistic regression | NR | NR | NR |
| Lu et al. [62] 2015 | Model1: GRS+ (age, sex, and BMI); Model2: GRS +Model1 + smoking, drinking, pulse rate, and education; Model3: GRS + Model2 + SBP and DBP | Logistic regression and Cox regression | Model1: C-statistic = 0.650 [0.637–0.663] (without GRS), 0.655 [0.642–0.668] (with GRS) Model 2: C-statistic = 0.683 [0.670–0.695] (without GRS), 0.687 [0.675–0.700] (with GRS) Model 3: C-statistic = 0.774 [0.763–0.785] (without GRS), 0.777 [0.766–0.787] (with GRS) | NR | NR |
| Zhang et al. [63] 2015 | Five latent factors extracted from 11 biomarkers (BMI, SBP, DBP, FBG, TG, HDL-C, Hb, HCT, WBC, LC, NGC): inflammatory factor, blood viscidity factor, insulin resistance factor, blood pressure factor, lipid resistance factor, and age | Cox regression | Derivation cohort: AUC = 0.755 [0.746–0.763] (men), AUC = 0.801 [0.792–0.810] (women) Validation cohort: AUC = 0.755 [0.746–0.763] (men), AUC = 0.800 [0.791–0.810] (women) | NR | Internal, 10-fold cross-validation |

NR, not reported; SBP, systolic blood pressure; DBP, diastolic blood pressure; BP, blood pressure; BMI, body mass index; CVD, cardiovascular disease; HDL, high-density lipoprotein; WC, waist circumference; DM, diabetes mellitus; WHR, waist to hip ratio; WHtR, waist to height ratio; ISH, isolated systolic hypertension; IDH, isolated diastolic hypertension; AUC, area under the curve; AROC, area under the receiver operating characteristic curve; LR, logistic regression; GEE, Generalized estimating equations; LOOCV, leave-one-out cross-validation: HL, Hosmer Lemeshow; GRS, genetic risk score; SNP, single-nucleotide polymorphism; CFS, correlation-based feature subset selection; FBG, fasting blood glucose; TG, triglycerides; HDL-C, high-density lipoprotein cholesterol; Hb, hemoglobin; HCT, hematocrit; WBC, white blood cell count; LC, lymphocyte count; NGC, neutrophil granulocyte count

**A.**

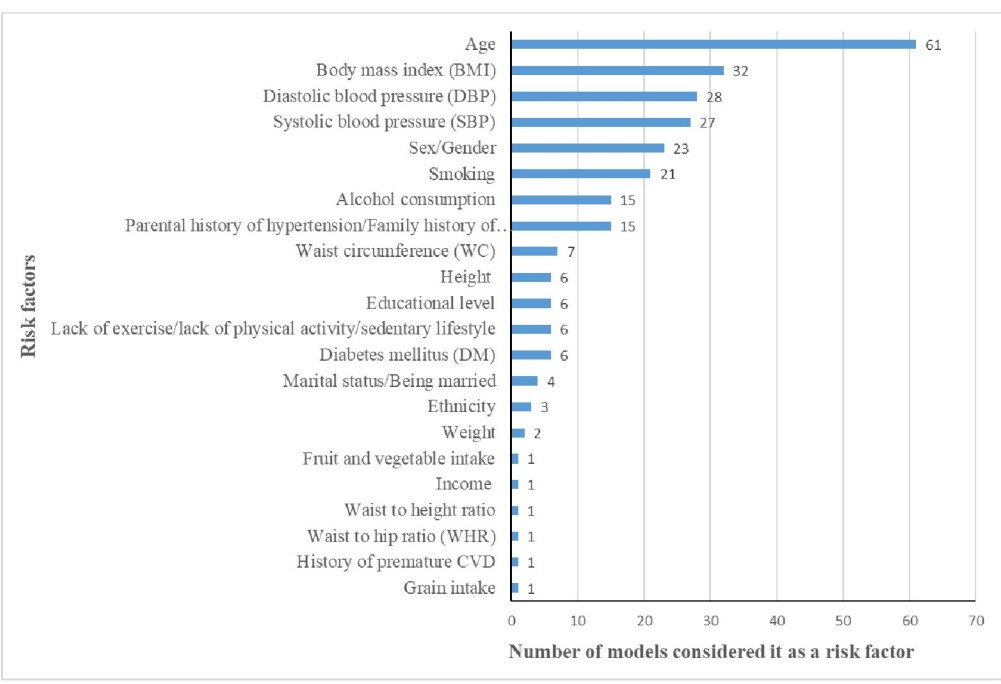

**B.**

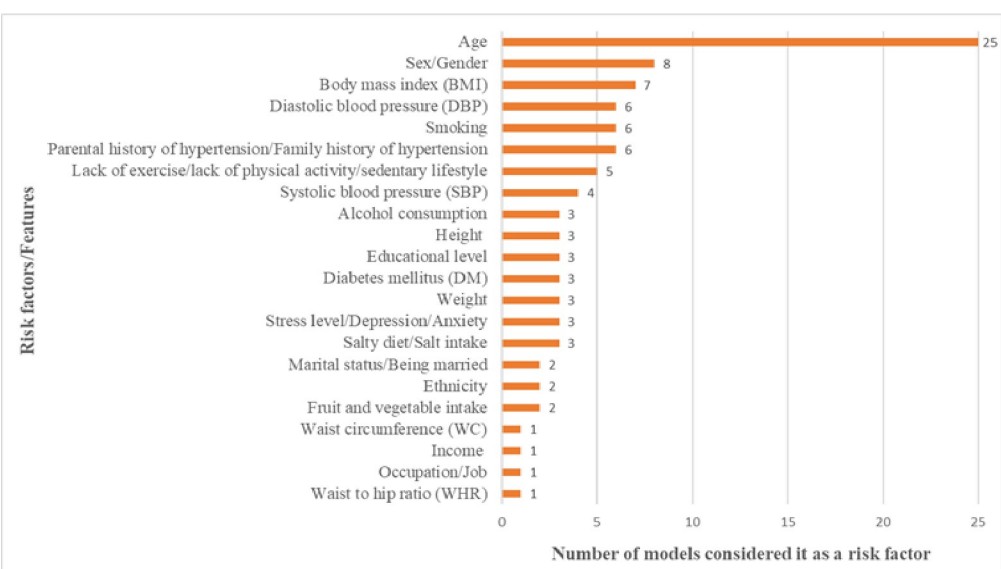

**Fig 2.** Conventional risk factors considered by traditional regression-based models (A) and by machine learning-based models (B).

Models were primarily developed either in white Caucasian or Asian populations. The number of risk factors/features considered to create the model ranged from 2 to 169, with a median of 7 risk factors per model. Age was the most common risk factor considered in 25 models, followed by sex/gender (8 models), BMI (7 models), DBP (6 models), smoking (6 models), and parental history of hypertension (6 models). The distribution of the conventional risk factors

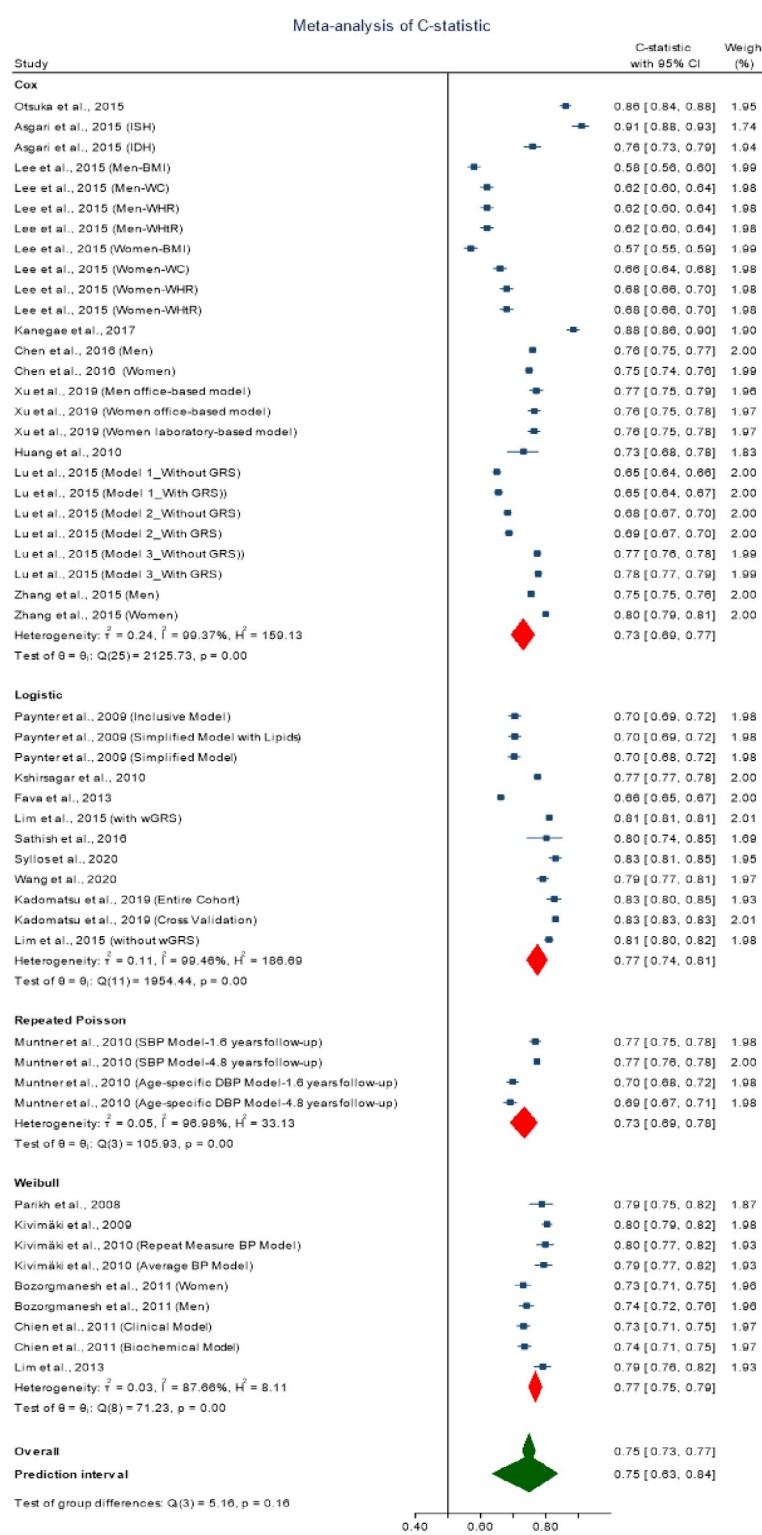

**Fig 3. Forest plot of traditional regression-based models with 95% prediction interval.**

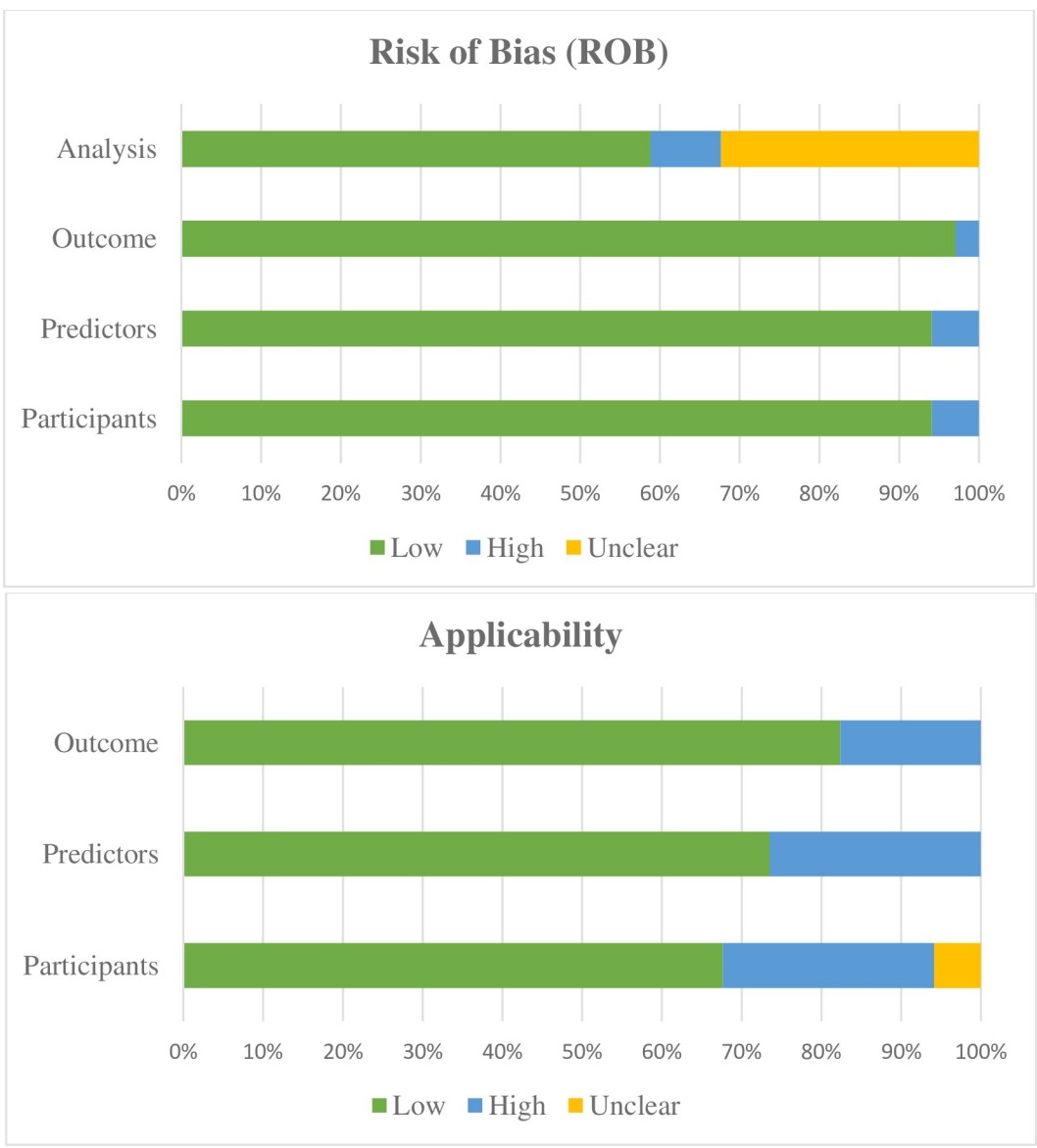

**Fig 4. Graphical summary presenting the percentage of hypertension risk prediction studies rated by level of concern, risk of bias (ROB), and applicability for each domain.**

considered in machine learning models is presented in Fig 2B. Hypertension was predominantly defined using SBP ≥ 140 mm Hg, DBP ≥ 90 mm Hg, or antihypertensive medication. Artificial neural network (ANN) was the most common method used to develop the models. Different studies reported different performance measures, and accuracy and AUC/C-statistic were the two most commonly reported measures. Most of the studies did not report calibration measures. In studies that reported discrimination, the AUC (or C-statistic) values range from 0.64 to 0.93.

## Meta-analysis of machine learning-based models

The overall pooled C-statistics of the machine learning-based models was 0.76 [0.72–0.79] with high heterogeneity in the discriminative performance of these models ($I^2$ = 99.9, Cochran

Table 3. Information about existing hypertension prediction models developed using machine learning algorithms from selected studies.

| Study | Data Location | Sample Size | Risk Factors Included | Outcome Considered | Definition of Outcome Predicted | Modeling Method Used | Performance Measure |
|---|---|---|---|---|---|---|---|
| Falk CT [64] 2003 | USA | 300 records each for training and validating | Seven input values: sex; age; total cholesterol; fasting glucose; fasting HDL; fasting triglycerides; body mass index (BMI) | High blood pressure | SBP > 140 mm Hg or DBP > 90 mm Hg | Two neural network programs: NNdriver and SNNS | Classification success rate. Training: 91%-98%, (Strategy 1), 70%-87% (Strategy 2); Validation: 59% (Strategy 1), 63% (Strategy 2) |
| Farran et al. [65] 2013 | Kuwait | 10,632 (6759 hypertensive and 3873 non-hypertensive) | BMI, age, ethnicity, and diagnosis for diabetes | Incident hypertension, type 2 diabetes, and comorbidity | NR | Logistic regression (LR), k-nearest neighbors, support vector machines, and multifactor dimensionality reduction (MDR) | Classification accuracy: 90% (hypertension) |
| Huang et al. [35] 2010 | China | Training: 2438, Validation: 616 | High educational level, predominantly sedentary work, positive family history of HTN, overweight, dysarteriotony, alcohol intake, salty diet, more vegetable and fruit intake, meat consumption, and regular physical exercise | Hypertension | Average SBP or DBP > 139 mmHg or > 89 mmHg, respectively | Logistic regression model (LRM) and artificial neural network (ANN) model (back-propagated delta rule networks) | AUC: 0.900 ± 0.014 (ANN model) AUC: 0.732 ± 0.026 (LRM) |
| Kwong et al. [66] 2018 | NR | 498 | Age, BMI, exercise level, alcohol consumption level, smoking status, stress level, and salt intake level | Systolic blood pressure (SBP) | BP readings > 140 mmHg | Two artificial neural networks (ANN): Back-propagation (BP) neural network and radial basis function (RBF) neural network validate the prediction system | Average Accuracy, BP ANN: 94.28% (male), 93.74% (female) RBF ANN: 91.06% (male), 90.44% (female) |
| Polak et al. [67] 2008 | USA | 159,989 records | High blood cholesterol, number of cigarettes smoked now, age, weight, height, sex | Hypertension | NR | Artificial neural network (ANN): Around 250 architectures of backpropagation (BP) and fuzzy networks | Classification rate and AUROC, different values for different Nets architecture |
| Priyadarshini et al. [68] 2018 | USA | NR | SBP, DBP, total cholesterol (TC), high-density lipoprotein (HDL), low-density lipoprotein (LDL), plasma glucose concentration (PGC), and heart rate (HR) | Hypertension attack | DBP or SBP > 90 mm Hg or > 120 mm Hg, respectively, for at least two measuring instances | Deep neural network model | Confusion/performance matrix formed out of four evaluating parameters: accuracy 88%, precision 92%, recall 82%, and F1 score 76% (average value over 20 iterations) |

(*Continued*)

**Table 3.** (Continued)

| Study | Data Location | Sample Size | Risk Factors Included | Outcome Considered | Definition of Outcome Predicted | Modeling Method Used | Performance Measure |
|---|---|---|---|---|---|---|---|
| Sakr et al. [36] 2018 | USA | 23,095 | Age, METS, resting systolic blood pressure, peak diastolic blood pressure, resting diastolic blood pressure, HX coronary artery disease, the reason for the test, history of diabetes, percentage HR achieved, race, history of hyperlipidemia, Aspirin use, hypertension response | Hypertension | NR | Six machine learning techniques: LogitBoost (LB), Bayesian network classifier (BN), locally weighted naïve Bayes (LWB), artificial neural network (ANN), support vector machine (SVM), and random tree forest (RTF) | AUC, F-Score, Sensitivity, Specificity, Precision, and RMSE. AUC (0.93), F-Score (86.70%), Sensitivity (69,96%) and Specificity (91.71%) for RTF model in 10-fold cross-validation AUC (0.88), Sensitivity (74.30%), Precision (73.50%), and F-Score (73.90%) for RTF model in holdout method |
| Tayefi et al. [69] 2017 | Iran | 9078 | Age, gender, BMI, marital status, level of education, occupation status, depression and anxiety status, physical activity level, smoking status, LDL, triglyceride, total cholesterol, fasting blood glucose, uric acid, and hs-CRP in Model 1 Age, gender, white blood cell, red blood cell, hemoglobin, hematocrit, mean corpuscular volume, mean corpuscular hemoglobin, platelets, red cell distribution width and platelet distribution width in Model 2 | Hypertension | SBP of 140 mm Hg, DBP of 90 mm Hg, and/or current use of antihypertensive drugs | Decision tree | Accuracy, sensitivity, specificity, and area under the ROC curve (AUC): For Model 1, the values are 73%, 63%, 77% and 0.72, respectively, and for Model 2 were 70%, 61%, 74% and 0.68, respectively |
| Wu et al. [70] 2015 | USA | 75 females and 165 males | Age, gender, serum cholesterol, fasting blood sugar and electrocardiographic signal, heart rate | Systolic blood pressure | SBP and DBP > 140 mm Hg and 90 mm Hg, respectively | Two neural network algorithms: back-propagation neural network and radial basis function network | The absolute difference (error) between the real value and predicted values |
| Wu et al. [71] 2016 | NR | 498 | Age, BMI, gender, exercise level, alcohol consumption, stress level, salt intake level, smoke status, cholesterol, and blood glucose | Systolic blood pressure | SBP > 140 mm Hg | Two artificial neural networks: back-propagation neural network and radial basis function neural network | The average prediction errors (absolute difference between the predicted value and measured value): 51.9% for men and 52.5% for women (backpropagation neural network) 51.8% for men and 49.9% for women (radial basis function network) |
| Ye et al. [37] 2018 | USA | 823,627 (training cohort/ retrospective cohort), 680,810 (validation cohort/ prospective cohort) | Total 169 features: 2 demographic features, 14 socioeconomic characteristics, 30 diagnostic diseases, 6 laboratory tests, 98 medication prescriptions, and 19 clinical utilization measures | Incident essential hypertension | ICD, 9th Revision, Clinical Modification (ICD-9-CM) diagnosis codes from category 401 | A supervised machine learning and data mining tool, XGBoost | AUC = 0.917 (retrospective cohort), AUC = 0.870 (prospective cohort) |

(*Continued*)

**Table 3.** (*Continued*)

| Study | Data Location | Sample Size | Risk Factors Included | Outcome Considered | Definition of Outcome Predicted | Modeling Method Used | Performance Measure |
|---|---|---|---|---|---|---|---|
| Zhang et al. [72] 2018 | NR | A total of 15,628,501 sets of valid characteristic attributes data | Seven input features: right atrium (AVR), left atrium (AVL), anterior atrium (AVF), photoplethysmography (PPG), oxygen saturation (SPO2), pulse transit time (PTT), heart rate (HR) | Blood pressure | NR | CART (classification and regression tree) model | Four evaluation indexes: accuracy rate, root mean square error (RMSE), deviation rate, and the Theil inequality coefficient (TIC) |
| Völzke et al. [31] 2013 | Germany | Training set: 803 Validation set: 802 External validation cohort: 2887 | Age, mean arterial pressure, rs16998073, serum glucose, and urinary albumin concentrations, the interaction between age and serum glucose, interaction between rs16998073 and urinary albumin concentrations | Incident hypertension | SBP $\geq$ 140 mmHg and DBP $\geq$ 90 mmHg | Bayesian network | Training set: AUC = 0.78 [0.74–0.82], Validation set: AUC = 0.79 [0.75–0.83], External validation set: AUC = 0.77 [0.74–0.80]; Training set: HL Chi-square = 11.82 (p = 0.16), Validation set: HL Chi-square = 11.65 (p = 0.17), External validation set: H-L Chi-square = 40.6 (p < 0.01) |
| Lee et al. [51] 2014 | Korea | 12,789 | Women: Height, age, neckC, axillaryC, ribC, waistC, pelvicC, rib_hip, waist_hip, pelvic_hip, rib_pelvic, axillary_rib, chest_rib, axillary_chest, forehead_neck (CFS), height, age, foreheadC, neckC, hipC, axillary_hip, axillary_pelvic, chest_pelvic, chest_rib (NB-wrapper) Men: Age, foreheadC, neckC, axillaryC, chestC, RibC, waistC, pelvicC, hipC, rib_hip, waist_hip, rib_pelvic, waist_pelvic, chest_waist, forehead_rib, chest_rib, axillary_chest, forehead_neck (CFS), height, age, foreheadC, neckC, axillaryC, hipC, rib_hip, pelvic_hip, neck_pelvic, waist_pelvic, chest_waist, chest_rib, neck_chest, forehead_neck (NB-wrapper) | Hypertension and hypotension | SBP $\geq$ 140 mmHg and/or DBP $\geq$ 90 mmHg or physician-diagnosed hypertension | Naive Bayes algorithm (NB) | Women: AUC = 0.696 (NB-CFS), 0.713 (NB-wrapper) Men: AUC = 0.64 (NB-CFS), 0.646 (NB-wrapper) |

(*Continued*)

**Table 3.** (*Continued*)

| Study | Data Location | Sample Size | Risk Factors Included | Outcome Considered | Definition of Outcome Predicted | Modeling Method Used | Performance Measure |
|---|---|---|---|---|---|---|---|
| Xu et al. [56] 2019 | China | 4796 | M1 Model: Age, SBP, DBP, hypertension parental history, WC, interaction item of age with WC, and interaction item of age with DBP<br>W1 Model: Age, SBP, DBP, WC, fruit and vegetable intake, hypertension parental history, interaction item of age with WC, and interaction of age with DBP | Hypertension | SBP $\geq$ 140 mm Hg and/or DBP $\geq$ 90 mm Hg and/or a diagnosis of hypertension by a physician and currently receiving anti-hypertension treatment | Artificial neural network (ANN), naive Bayes classifier (NBC), and classification and regression tree (CART) | Testing Set Men: AUC = 0.773 [0.752–0.793] (ANN), 0.760 [0.738–0.781] (NBC), 0.722 [0.699–0.743] (CART)<br>Testing Set Women: AUC = 0.756 [0.737–0.775] (ANN), 0.761 [0.742–0.779] (NBC), 0.698 [0.677–0.717] (CART)<br>Testing Set Men: Modified Nam-D'Agostino test Chi-square = 29.274, p = 0.0006 (ANN); 82.269, p < 0.00001 (NBC); 5.249, p = 0.072 (CART)<br>Testing Set women: Modified Nam-D'Agostino test Chi-square = 4.744, p = 0.314 (ANN); 189.754, p < 0.00001 (NBC); 19.733, p = 0.00005 (CART) |
| Wang et al. [57] 2015 | USA | 308,711 | Exercise, diabetes, hyperlipemia, age, marriage, education, income, weight, height, sex, smoke, drink | Hypertension | NR | Multi-layer perception neural network | Accuracy, sensitivity, specificity, and AUC. Average AUC = 0.77 with h vary from 8 to 11 (neural network); Accuracy = 72% (neural network) |
| Ture et al. [59] 2005 | Turkey | 694 | Age, sex, family history of hypertension, smoking habits, lipoprotein (a), triglyceride, uric acid, total cholesterol, and BMI | Essential hypertension | The average of 3 or more DBP measurements on at least 3 subsequent visits is $\geq$ 90 mmHg, or when the average of multiple SBP readings on 3 or more subsequent visits is consistently $\geq$ 140 mmHg | Three decision trees (Chi-squared automatic interaction detector. Classification and regression tree, quick, unbiased, efficient statistical tree); two neural networks (multi-layer perceptron, radial basis function) | Sensitivity, specificity, and predictive rate (PR). Values not reported. |
| Zhao et al. [73] 2008 | China/Asians | Total: 4759 (2411 hypertensive and 2,348 age-matched and sex-matched healthy controls) | MDR Model: 4-locus model consisted of the SNP KCNMB1-rs11739136, RGS2-rs34717272, PRKG1-rs1881597, and MYLK-rs36025624; CART Model: RGS2, PRKG1, KCNMB1, and MYLK | Hypertension CHECK | Average SBP $\geq$ 150 mm Hg, an average DBP $\geq$ 95 mm Hg, or current use of antihypertensive medication | Multifactor-dimensionality reduction (MDR) and classification and regression trees (CART) | MDR Model: Accuracy = 52.98%, cross-validation consistency = 9.7 |

(*Continued*)

**Table 3.** (Continued)

| Study | Data Location | Sample Size | Risk Factors Included | Outcome Considered | Definition of Outcome Predicted | Modeling Method Used | Performance Measure |
|-------|---------------|-------------|-----------------------|--------------------|-----------------------------------|----------------------|---------------------|
| Wang et al. [57] 2014 | China/ Asians | 1009 hypertensive patients and 756 normotensive controls | Genes | Hypertension | Mean SBP ≥ 140 mmHg and/or DBP ≥ 90 mmHg on two occasions and/or the current usage of antihypertensive drug treatment | Multifactor dimensionality reduction (MDR) model | The best MDR model testing accuracy = 0.6331, cross-validation consistency = 10 |
| Zhao et al. [74] 2014 | China/ Asians | 1009 hypertensive patients and 756 normotensive controls | The best MDR model included rs5804 and BMI | Hypertension | Mean SBP of at least 140 mmHg or a mean DBP of at least 90 mmHg or the current intake of antihypertensive drugs | Multifactor dimensionality reduction (MDR) model | The best MDR model: testing accuracy of 0.7309 and a maximum cross-validation consistency of 10 (P < 0.001) |

ICD, international classification of diseases

Q-statistic p < 0.001) (Fig 5). Like traditional regression-based models, we did not perform stratified pooled results by modeling type due to diversity in the modeling method. The 95% approximate prediction interval for the overall C-statistics was from 0.63 to 0.84 (Fig 5).

We explored possible sources of heterogeneity in the overall pooled C-statistics through meta-regression as before. However, meta-regression did not identify any of age of the participants (p = 0.358), the number of risk factors considered in the model (p = 0.812), sex of the participants, that is being male compared to female (p = 0.886) and both male and female compared to female-only (p = 0.787), sample size considered in the model (p = 0.577), or ethnicity of the study participants (p = 0.326) as the potential source of high heterogeneity in the C-statistic.

## Study characteristics of externally validated models

Only four models [22, 30–32] were found to be externally validated in a different population. Detailed characteristics of the studies that validated these four models are presented in S3 Table. The Framingham hypertension risk model (FHRS) is the only validated model in more than one external population. The FHRS [22] model was validated by eight different studies in diverse populations of 122,348 participants. Study participants had an age range of 18 to 84 years with follow-up time (mean/median/total) from 1.6 years to 25 years. Almost all studies reported performance measures of the FHRS. The Hosmer-Lemeshow test was used to report calibration, while the C-statistic (or AUC) was used to report discrimination. The values of the reported C-statistic ranged from 0.54 to 0.84. Models by Lim et al. [30], Völzke et al. [31], and Kanegae et al. [32] were validated only once in an external population by the same authors. Within these three models, performances were best for the model by Kanegae et al. [32], with a C-statistic of 0.85 [0.76–0.91].

## Meta-analysis of externally validated models

The pooled C-statistic of the FHRS [22] model was 0.75 [0.68–0.80] with high heterogeneity in the discriminative performance of this model ($I^2$ = 99.6, Cochran Q-statistic p < 0.001) (S3 Fig). The 95% approximate prediction interval for the C-statistic in the FHRS [22] was from 0.47 to 0.91 (S3 Fig). As the other three models were externally validated only once, pooling their performance measure was irrelevant.

## Meta-analysis of C-statistic

| Study | | C-statistic with 95% CI | Weight (%) |
|---|---|---|---|
| Huang et al., 2010 | | 0.90 [ 0.86, 0.93] | 3.14 |
| Sakr et al., 2018 (ANN_10-fold CV) | | 0.67 [ 0.66, 0.68] | 3.48 |
| Sakr et al., 2018 (LB_10-fold CV) | | 0.69 [ 0.68, 0.70] | 3.48 |
| Sakr et al., 2018 (LWB_10-fold CV) | | 0.67 [ 0.66, 0.68] | 3.48 |
| Sakr et al., 2018 (RTF_10-fold CV) | | 0.93 [ 0.93, 0.93] | 3.47 |
| Sakr et al., 2018 (BN_10-fold CV) | | 0.70 [ 0.69, 0.71] | 3.47 |
| Sakr et al., 2018 (SVM_10-fold CV) | | 0.71 [ 0.70, 0.72] | 3.47 |
| Sakr et al., 2018 (ANN_HO) | | 0.74 [ 0.73, 0.75] | 3.47 |
| Sakr et al., 2018 (LB_HO) | | 0.70 [ 0.69, 0.71] | 3.47 |
| Sakr et al., 2018 (LWB_HO) | | 0.70 [ 0.69, 0.71] | 3.47 |
| Sakr et al., 2018 (RTF_HO) | | 0.89 [ 0.88, 0.89] | 3.47 |
| Sakr et al., 2018 (BN_HO) | | 0.72 [ 0.71, 0.73] | 3.47 |
| Sakr et al., 2018 (SVM_HO) | | 0.59 [ 0.58, 0.60] | 3.48 |
| Tayefi et al., 2016 (Model 1) | | 0.72 [ 0.70, 0.74] | 3.45 |
| Tayefi et al., 2016 (Model 2) | | 0.68 [ 0.66, 0.70] | 3.45 |
| Ye et al., 2018 (Retrospective) | | 0.92 [ 0.92, 0.92] | 3.48 |
| Ye et al., 2018 (Prospective) | | 0.87 [ 0.87, 0.87] | 3.48 |
| Völzke et al., 2013 (Validation) | | 0.79 [ 0.75, 0.83] | 3.32 |
| Lee et al., 2014 (Women_NB-CFS) | | 0.70 [ 0.68, 0.71] | 3.46 |
| Lee et al., 2014 (Women_NB-wrapper) | | 0.71 [ 0.70, 0.73] | 3.46 |
| Lee et al., 2014 (Men_NB-CFS) | | 0.64 [ 0.62, 0.66] | 3.46 |
| Lee et al., 2014 (Men_NB-wrapper) | | 0.65 [ 0.63, 0.66] | 3.46 |
| Xu et al., 2019 (Men_ANN) | | 0.77 [ 0.75, 0.79] | 3.44 |
| Xu et al., 2019 (Men_NBC) | | 0.76 [ 0.74, 0.78] | 3.44 |
| Xu et al., 2019 (Men_CART) | | 0.72 [ 0.70, 0.74] | 3.44 |
| Xu et al., 2019 (Women_ANN) | | 0.76 [ 0.74, 0.77] | 3.45 |
| Xu et al., 2019 (Women_NBC) | | 0.76 [ 0.74, 0.78] | 3.45 |
| Xu et al., 2019 (Women_CART) | | 0.70 [ 0.68, 0.72] | 3.45 |
| Wang et al., 2015 | | 0.77 [ 0.77, 0.77] | 3.48 |
| **Overall** | | 0.76 [ 0.72, 0.79] | |
| **Prediction interval** | | 0.75 [ 0.63, 0.84] | |

0.40    0.60    0.80

Random-effects REML model

**Fig 5. Forest plot of machine regression-based models with 95% prediction interval.**

We explored possible sources of heterogeneity in the pooled C-statistics through meta-regression, and only the ethnicity (Whites versus Asians) of the study participants (p = 0.044) was identified as a source of high heterogeneity in the C-statistic of the FHRS model [22].

### Models developed using genetic risk factors/biomarkers

Genetic risk factors/biomarkers often contribute significantly to developing hypertension, and models were developed considering both conventional risk factors and biomarkers. In addition, there were models where biomarkers were used primarily in model building. Information about models developed using biomarkers (e.g., genetic risk scores) is presented in S4 Table. There were 11 studies where genetic risk factors/biomarkers were used in model building. Biomarkers are often considered very important for increasing the predictive performance of models. However, the pooled predictive performance (C-statistic) of the models that considered biomarkers primarily was 0.76 [0.71–0.80] (S4 Fig) and did not show an overall improvement in the models' predictive performance. Including genetic factors/biomarkers in the model has some drawbacks. Because information on those biomarkers is frequently unavailable and interpreting the models becomes difficult, the models become less suitable for daily clinical practice.

## Discussion

Many hypertension risk prediction models with reasonable predictive performance were identified in this systematic review, but only a few had external validation. Bias and applicability were noted as major concerns in many studies. Overall, there was little difference in the predictive performance of traditional statistical and machine learning models. Our findings are expanded on in the sections that follow.

The models were developed mostly in Caucasian or Asian populations. Because certain ethnic groups are more prone to hypertension (e.g., people of African descent [33]), research should include a diverse range of patients to create hypertension risk prediction models. Most of the traditionally developed models considered conventional risk factors for hypertension, which are readily available in clinical practice. Some models also used genetic risk factors, although the inclusion of genetic risk factors into the model did not improve the overall predictive performance of the models. The pooled analysis identified the overall predictive performance of the traditional regression-based models was good but with high heterogeneity. Stratified analysis by modeling methodology (e.g., logistic, Cox) within traditional regression-based models did not show much difference in predictive performance, and heterogeneity was still observed within the modeling methodology. The traditional models we identified in our search were mostly internally validated, often considered not enough for models' generalizability [34]. The FHRS [22] was the only model that had multiple external validations and good/acceptable pooled predictive performance. However, because the FHRS [22] showed high heterogeneity in its predictive performance, with ethnicity serving as a source of heterogeneity, and the model was built predominantly in a White population, we must proceed with caution when applying it to a completely different population. Models that have only single, or no validation need external validation, preferably by a different group of investigators, to guarantee the model's generalizability to a different population. Only a few traditional models were converted into risk score after their development. Presenting the risk derived from the model through scoring instead of a complex mathematical formula may facilitate the use of prediction models and subsequently improve the uptake of prediction models in clinical practice. The risk of bias (ROB) was "high" or "unclear" in a large portion of traditional model studies. This is primarily because many studies failed to meet the criteria in the "analysis" domain of ROB. In many studies, the applicability of the models was rated as "high concern" or "unclear concern" due to a failure to properly fulfil the "participants" criteria. Several models were developed in a specific population, making the models less applicable to the general adult population.

Since machine learning tools are more recent, advanced, and have a reputation for producing more accurate predictive performance, we assumed that models developed with these tools would outperform traditional regression-based models. However, we did not notice much difference in predictive performance between these two types of models. A few machine learning-based models (e.g., models by Huang et al. [35], Sakr et al. [36], and Ye et al. [37]) showed excellent discriminative performance; however, none of these models has ever been externally validated in an entirely different new population. In fact, none of the machine learning-based models have been externally validated. Consequently, the performance of those models in a new setting/population is quite uncertain. We also noticed high heterogeneity in the predictive performance (C-statistic) of machine learning models. Meta-regression using potential sources of heterogeneity failed to identify the real source of heterogeneity. One possible explanation is a difference in the methodology used to develop the machine learning-based models. Due to the various methods considered in different models, we were unable to investigate this potential source. We did not notice higher expected variability in machine learning-based models' future predictive performance compared to traditional regression-based models, as the 95% prediction interval for machine learning-based models was similar to traditional regression-based models.

We did not find any studies in this review that assessed the impact of adopting hypertension risk prediction models in clinical settings. Ideally, a prediction model, regardless of its development, should have an impact study to assess whether it improves clinical decision-making and patient health outcomes [5, 38].

There were two previous reviews on a similar topic where hypertension risk prediction models were identified through a systematic search and described their characteristics. Our review is different from previous studies and contributes to information on the prediction of hypertension risk and the identification of associated risk factors in the following ways: 1) we synthesized performance of the prediction models through meta-analysis and explored potential sources of heterogeneity; 2) we compared the performance of the prediction models developed using traditional statistical regression-based models and more recent machine learning-based models; 3) we provided a thorough evaluation of the quality of the studies among traditionally developed regression-based models; and 4) we described several additional models that have recently been derived.

One of our study's strengths is the extent of the systematic search, which includes four different databases, grey literature, and extensive use of the reference lists of the identified studies. To the best of our knowledge, this is the first study where a meta-analysis of predictive performance, together with assessment of heterogeneity, comparison of the predictive performance of traditional regression based-models and machine learning-based models, and a detailed critical appraisal of studies in hypertension risk prediction models has been performed. Nevertheless, our study also has limitations. We excluded non-English and non-French publications. While it is widely perceived that the English language is the primary language of science, the choice of scientific results in a particular language can incorporate language bias and may lead to incorrect conclusions [39]. We were only able to use C-statistics to compare the model performance, which could be insensitive to distinguish a model's ability to correctly stratify patients into clinically relevant risk groups [39, 40]. Calibration was quantified by different measures, and different studies often reported different calibration measures. This led to difficulty in synthesizing calibration measures through meta-analysis. A meta-analysis of calibration measures (e.g., O/E ratio) along with C-statistics could provide a comprehensive summary of the performance of these models [19]. Failing to assess publication bias amongst the studies is another potential limitation of this study. Recent guidelines [19] did not emphasize the need to assess publication bias for prediction model performance, which encouraged

us not to do so. Although studies have considered publication bias in a similar scenario before, we believe existing traditional publication bias assessment tools (e.g., funnel plot, Egger's test, Begg's test) are more appropriate for studies assessing statistically significant results (e.g., randomized controlled trial (RCT)) than studies assessing predictive performance (e.g., C-statistic) of the prognostic models. Instead, we assessed ROB using the PROBAST checklist. We also could not appraise studies that use machine learning algorithms to predict hypertension. Although most of the PROBAST signaling questions also apply to appraise machine learning algorithms, additional signaling questions are recommended to add due to differences in data analysis methods for machine learning algorithms and regression-based models [14, 15]. Machine learning algorithms use different variable selection strategies, different estimation techniques for variable–outcome estimations, and different ways to adjust for overfitting [14, 15]. When additional questions are added to the PROBAST, these questions need to be appropriately phrased, and specific guidance on assessing these signaling questions also needs to be provided [14, 15]. Considering these additional works, we refrain from appraising studies considered machine learning algorithms. Finally, despite our attempt to capture potential sources of heterogeneity in our study, we asked readers to be cautious while interpreting our findings as there may be a potential bias in our findings due to a limited number of studies included in the analysis and the study's failure to incorporate additional potential sources of bias in the analysis.

In summary, we attempted to provide a comprehensive evaluation of hypertension risk prediction models. We identified many models with acceptable-to-good predictive performance. We did not notice significant differences in the predictive performance of traditional regression-based models and machine learning-based models. Including genetic risk factors/biomarkers also did not show much improvement in the models' predictive performance. The quality of the studies was reasonable, with areas where further improvement is needed. Only a few of the multiple models developed had been externally validated, which is a concern. Also, there is a lack of impact studies. Models with external validation and impact studies are required to implement a prediction model in a clinical practice guideline. A model with accurate prediction is not beneficial if it is not generalizable to a different population or improves clinical decision-making and patient health outcomes.

## Supporting information

**S1 Checklist. PRISMA 2020 checklist.**
(DOCX)

**S1 Fig. The number of PROBAST criteria satisfied by different studies.**
(DOC)

**S2 Fig. Response to different signaling questions by the number of studies.**
(DOC)

**S3 Fig. Forest plot of externally validated models with 95% prediction interval.**
(DOC)

**S4 Fig. Forest plot of models primarily developed using genetic risk factors/biomarkers with a 95% prediction interval.**
(DOC)

**S1 Table. Keywords used to search in MEDLINE.**
(DOC)

**S2 Table. Study quality assessment using PROBAST.**
(DOC)

**S3 Table. Information about external validation studies of existing traditional hypertension prediction models from selected studies.**
(DOC)

**S4 Table. Information about existing hypertension prediction models developed using biomarkers (genetic risk score) from the selected studies.**
(DOC)

## Author Contributions

**Conceptualization:** Mohammad Ziaul Islam Chowdhury.

**Data curation:** Mohammad Ziaul Islam Chowdhury, Iffat Naeem.

**Formal analysis:** Mohammad Ziaul Islam Chowdhury.

**Investigation:** Mohammad Ziaul Islam Chowdhury, Iffat Naeem, Hude Quan, Alexander A. Leung, Khokan C. Sikdar, Maeve O'Beirne.

**Methodology:** Mohammad Ziaul Islam Chowdhury, Hude Quan, Alexander A. Leung, Tanvir C. Turin.

**Software:** Mohammad Ziaul Islam Chowdhury.

**Supervision:** Hude Quan, Alexander A. Leung, Khokan C. Sikdar, Maeve O'Beirne, Tanvir C. Turin.

**Validation:** Mohammad Ziaul Islam Chowdhury, Iffat Naeem.

**Visualization:** Mohammad Ziaul Islam Chowdhury.

**Writing – original draft:** Mohammad Ziaul Islam Chowdhury.

**Writing – review & editing:** Mohammad Ziaul Islam Chowdhury, Alexander A. Leung.

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
