## [Decision Letter · Decision Letter 0]

8 Nov 2021

PONE-D-21-31564Prediction of hypertension using traditional regression and machine learning models: A systematic review and meta-analysisPLOS ONE

Dear Dr. Turin,

Thank you for submitting your manuscript to PLOS ONE. After careful consideration, we feel that it has merit but does not fully meet PLOS ONE’s publication criteria as it currently stands. Therefore, we invite you to submit a revised version of the manuscript that addresses the points raised during the review process.

We look forward to receiving your revised manuscript.

Kind regards,

Antonio Palazón-Bru, PhD

Academic Editor

PLOS ONE

Journal Requirements:

Reviewers' comments:

Reviewer's Responses to Questions

**Comments to the Author**

1. Is the manuscript technically sound, and do the data support the conclusions?

Reviewer #1: Yes

Reviewer #2: Partly

2. Has the statistical analysis been performed appropriately and rigorously? 

Reviewer #1: Yes

Reviewer #2: Yes

3. Have the authors made all data underlying the findings in their manuscript fully available?

Reviewer #1: Yes

Reviewer #2: Yes

4. Is the manuscript presented in an intelligible fashion and written in standard English?

Reviewer #1: Yes

Reviewer #2: No

5. Review Comments to the Author

Reviewer #1: The authors compared the predictive performance of two types of hypertension risk prediction models: those developed using traditional regression-based and those using machine learning approaches. They searched the MEDLINE, EMBASE, Web of Science, Scopus, and the grey literature for studies predicting the risk of hypertension among the general adult population. They used the C-statistic, and a random-effects meta-analysis was used to obtain pooled estimates from the individual studies The potential sources of heterogeneity was assessed using meta-regression, and study quality was assessed using the PROBAST (Prediction model Risk Of Bias ASsessment Tool) checklist. They selected 52 articles for systematic review and 32 for meta-analysis out of the 14,778 citations that they retrieved. They observed modest and similar overall pooled C-statistics of 0.75 [0.73 – 0.77] for the traditional regression-based models and 0.76 [0.72 – 0.79] for the machine learning-based models. There was high heterogeneity in the C-statistic in both methods. The age (p = 0.011), and sex (p = 0.044) of the participants and the number of risk factors considered in the model (p = 0.001) were identified as sources of heterogeneity in traditional regression-based models. The authors concluded that only a few models were externally validated, that the risk of bias and applicability was a concern in many studies that many models with acceptable-to-good predictive performance were identified that overall discrimination was similar between models derived from traditional regression analysis and machine learning methods and that external validation and of the hypertension risk prediction model in clinical practice are required.

The authors may wish to consider the following.

1. Selecting a small number of studies may have led to biased conclusions.

2. The variability in the duration of follow-up time (1.6 years to 30 years), the age of the participants (15 to 90 years), SBP ≥ 140 mm Hg, DBP ≥ 90 mm Hg, or SBP ≥ 130 mm Hg, DBP ≥ 80 mm Hg, and or use of antihypertensive medication may have led to biased conclusions.

3. In addition, the variability on the geographic region, time, or gender of the study participants may have led to biased conclusions.

4. The authors may wish to expand the limitations section of the Discussion in page 18 to include items 1, 2 and 3 above.

5. Would the authors agree to include the last sentence of the manuscript “we attempted to provide a comprehensive evaluation of hypertension risk prediction models” in the Abstract?

Reviewer #2: My review is attached as a document for ease of reading., but I also include it here:

Review: Chowdhury et al “Prediction of hypertension using traditional regression and machine learning models: A systematic review and meta-analysis”

Overview

In this paper Chowdhury et al provide a systematic review and meta-analysis comparing prediction models for the development of hypertension in the general population derived using traditional regression-based and machine learning approaches.

Meta-analysis was only possible for measures of discrimination. Overall the pooled c-statistics on meta-analysis are similar and of moderate-good performance between traditional regression-based and machine learning derived models. High heterogeneity was found, with sources identified for traditional regression-based models through meta-regression. Only one model has been extensively externally validated (Framingham Hypertension risk model) but it showed significant heterogeneity in meta-analysis. Performance of risk models for hypertension have only been appropriately checked in Asian and Caucasian populations and clinical implementation has not been assessed.

Overall impression

I would like to congratulate the reviewers on an extremely thorough and methodologically sound systematic review and meta-analysis. My main concerns relate to the structure and writing of the discussion section, and the presentation the table.

Major issues

• The aims of the study are clearly delineated in the introduction (point 1-4). However I do not feel the structure of the discussion follows these aims or highlights the most salient findings of the analysis. Furthermore in my opinion the discussion section is too long. It would be better presented:

o Major findings of the study (3-4 points)

o Discussion of previous literature and how this differs

o Future areas for research / gaps in knowledge

o Limitations

o Final conclusion

• The presentation of table 1 is extremely difficult to follow. The presentation of so many columns means that some of the entries for each study take up an entire page. It would be better to break this up into at least 2/3 tables e.g. between study population characteristics, model development characteristics/performance, variables used in model; and all these tables do not need to be in the main file (eg Himmelreich et al -> https://academic.oup.com/europace/article/22/5/684/5721485)

• Why are traditional regression model study characteristics included in main paper but not machine learning counterparts. It would be better to present them more equally

• There are wide prediction intervals suggesting significant heterogeneity. Have you considered a Bayesian approach for meta-analysis? Frequentist methods can produce prediction intervals with poor coverage when there is a mixture of study sizes (https://pubmed.ncbi.nlm.nih.gov/30032705/)

Minor issues

• I note some of the models for predicting hypertension use systolic blood pressure and diastolic blood pressure. Does this not appear ‘double-dipping’ to include a variable that may well be an outcome? Does this not require some comment?

• Page 13 line 330 – please be more specific than ‘basically’

• Page 15 line 388 – I belive it should be ‘models’

• Page 17 line 446-447 does not make sense

• Figure 1 – I believe the reasons for exclusion would be better ordered alphabetically or in descending number of records excluded

6. PLOS authors have the option to publish the peer review history of their article (what does this mean?). If published, this will include your full peer review and any attached files.

Reviewer #1: **Yes: **John B. Kostis

Reviewer #2: No

---

## [Author Response · Author response to Decision Letter 0]

8 Mar 2022

Response to journal requirements and reviewers’ comments

Journal Requirements:

RESPONSE: Thank you. We have revised our manuscript accordingly.

RESPONSE: Thank you. None of the authors received any funding for this study. We have now stated, “The authors received no specific funding for this work” in our revised manuscript and in the cover letter.

RESPONSE: Thank you. Since our study is a systematic review and we did not use any primary data in our analysis, we have now revised our data availability statement as follows: “All relevant data are within the manuscript and its Supporting information files”. We have included this statement in our revised manuscript and in the cover letter.

RESPONSE: Thank you. Yes, we would like to make changes to our Data Availability statement. Since our study is a systematic review and we did not use any primary data in our analysis, we have now revised our data availability statement as follows: “All relevant data are within the manuscript and its Supporting information files”. We have included this statement in our revised manuscript and in the cover letter.

RESPONSE: Thank you. We have now included captions for Supporting Information files at the end of our manuscript.

Reviewers' comments:

Reviewer's Responses to Questions

Comments to the Author

1. Is the manuscript technically sound, and do the data support the conclusions?

Reviewer #1: Yes

Reviewer #2: Partly

2. Has the statistical analysis been performed appropriately and rigorously?

Reviewer #1: Yes

Reviewer #2: Yes

3. Have the authors made all data underlying the findings in their manuscript fully available?

Reviewer #1: Yes

Reviewer #2: Yes

4. Is the manuscript presented in an intelligible fashion and written in standard English?

Reviewer #1: Yes

Reviewer #2: No

5. Review Comments to the Author

Reviewer #1: 

COMMENT. The authors compared the predictive performance of two types of hypertension risk prediction models: those developed using traditional regression-based and those using machine learning approaches. They searched the MEDLINE, EMBASE, Web of Science, Scopus, and the grey literature for studies predicting the risk of hypertension among the general adult population. They used the C-statistic, and a random-effects meta-analysis was used to obtain pooled estimates from the individual studies The potential sources of heterogeneity was assessed using meta-regression, and study quality was assessed using the PROBAST (Prediction model Risk Of Bias ASsessment Tool) checklist. They selected 52 articles for systematic review and 32 for meta-analysis out of the 14,778 citations that they retrieved. They observed modest and similar overall pooled C-statistics of 0.75 [0.73 – 0.77] for the traditional regression-based models and 0.76 [0.72 – 0.79] for the machine learning-based models. There was high heterogeneity in the C-statistic in both methods. The age (p = 0.011), and sex (p = 0.044) of the participants and the number of risk factors considered in the model (p = 0.001) were identified as sources of heterogeneity in traditional regression-based models. The authors concluded that only a few models were externally validated, that the risk of bias and applicability was a concern in many studies that many models with acceptable-to-good predictive performance were identified that overall discrimination was similar between models derived from traditional regression analysis and machine learning methods and that external validation and of the hypertension risk prediction model in clinical practice are required.

RESPONSE: Thank you so much for your excellent comment.

COMMENT. The authors may wish to consider the following.

1. Selecting a small number of studies may have led to biased conclusions.

2. The variability in the duration of follow-up time (1.6 years to 30 years), the age of the participants (15 to 90 years), SBP ≥ 140 mm Hg, DBP ≥ 90 mm Hg, or SBP ≥ 130 mm Hg, DBP ≥ 80 mm Hg, and or use of antihypertensive medication may have led to biased conclusions.

3. In addition, the variability on the geographic region, time, or gender of the study participants may have led to biased conclusions.

4. The authors may wish to expand the limitations section of the Discussion in page 18 to include items 1, 2 and 3 above.

RESPONSE: Thank you so much for your excellent comments. We agree with the reviewer that items 1, 2, and 3 could be potential sources of bias. However, we would like to point out here that we considered most of those listed items as potential sources of heterogeneity in C-statistics in our analysis. For example, age, gender (sex), the definition of hypertension used (the cut-off level used to define hypertension as the reviewer indicated), and ethnicity (which reflected the influence of geographic region) were considered as the potential sources of heterogeneity in the C-statistics in our analysis. However, we acknowledge that variations on these items may lead to biased conclusions in study findings, and we have included these as limitations in our revised manuscript. The following lines were added to the revised manuscript: 

“Finally, despite our attempt to capture potential sources of heterogeneity in our study, we asked readers to be cautious while interpreting our findings as there may be a potential bias in our findings due to a limited number of studies included in the analysis and the study's failure to incorporate additional potential sources of bias in the analysis.”

Please see Page 18, lines 461-464 in the revised manuscript.

COMMENT. 5. Would the authors agree to include the last sentence of the manuscript “we attempted to provide a comprehensive evaluation of hypertension risk prediction models” in the Abstract?

RESPONSE: Thank you. We have included this sentence in the abstract.

Please see Page 3, lines 73-74 in the revised manuscript.

Reviewer #2: My review is attached as a document for ease of reading., but I also include it here:

Review: Chowdhury et al “Prediction of hypertension using traditional regression and machine learning models: A systematic review and meta-analysis”

COMMENT. Overview

In this paper Chowdhury et al provide a systematic review and meta-analysis comparing prediction models for the development of hypertension in the general population derived using traditional regression-based and machine learning approaches.

Meta-analysis was only possible for measures of discrimination. Overall the pooled c-statistics on meta-analysis are similar and of moderate-good performance between traditional regression-based and machine learning derived models. High heterogeneity was found, with sources identified for traditional regression-based models through meta-regression. Only one model has been extensively externally validated (Framingham Hypertension risk model) but it showed significant heterogeneity in meta-analysis. Performance of risk models for hypertension have only been appropriately checked in Asian and Caucasian populations and clinical implementation has not been assessed.

Overall impression

I would like to congratulate the reviewers on an extremely thorough and methodologically sound systematic review and meta-analysis. My main concerns relate to the structure and writing of the discussion section, and the presentation the table.

RESPONSE: Thank you so much for your comments and suggestions

COMMENT. Major issues

• The aims of the study are clearly delineated in the introduction (point 1-4). However I do not feel the structure of the discussion follows these aims or highlights the most salient findings of the analysis. Furthermore in my opinion the discussion section is too long. It would be better presented:

o Major findings of the study (3-4 points)

o Discussion of previous literature and how this differs

o Future areas for research / gaps in knowledge

o Limitations

o Final conclusion

RESPONSE: Thank you so much for taking the time to make such an insightful observation. It is true that the discussion portion is overly lengthy, as stated by the reviewer. However, we would want to point out that our objective was to provide a full explanation of the existing hypertension risk prediction models, which we have done. We discovered 117 models that are extremely huge as a result of our search and addressing the primary conclusions of these models took up a significant amount of space in the discussion section. We hope that offering a full discussion will assist readers in understanding the silent characteristics of the models that have been found.

We appreciate your suggestions for the layout of the discussion part, and we acknowledge that we have made every effort to provide the discussion sections in the suggested manner. In addition, we have reduced the length of the discussion part by deleting redundant content whenever possible, as indicated by the reviewer. Please see the revised discussion section.

Please see Pages 15-19, lines 374-474 in the revised manuscript.

COMMENT. • The presentation of table 1 is extremely difficult to follow. The presentation of so many columns means that some of the entries for each study take up an entire page. It would be better to break this up into at least 2/3 tables e.g. between study population characteristics, model development characteristics/performance, variables used in model; and all these tables do not need to be in the main file (eg Himmelreich et al -> https://academic.oup.com/europace/article/22/5/684/5721485)

RESPONSE: Thank you for your comment. We agree with the reviewer. As per the reviewer’s suggestion, we have now split the information in Table 1 into two tables, Table 1 and Table 2. Please see Table 1 and Table 2 in the revised manuscript. Pages 32 – 44.

COMMENT. • Why are traditional regression model study characteristics included in main paper but not machine learning counterparts. It would be better to present them more equally.

RESPONSE: Thank you for your comment. We agree with the reviewer. As per the reviewer’s suggestion, we have now added the study characteristics of the machine learning models in the main paper. Please see the newly added Table 3 in the revised manuscript. Pages 45- 52.

COMMENT. • There are wide prediction intervals suggesting significant heterogeneity. Have you considered a Bayesian approach for meta-analysis? Frequentist methods can produce prediction intervals with poor coverage when there is a mixture of study sizes (https://pubmed.ncbi.nlm.nih.gov/30032705/)

RESPONSE: Thank you for making such an astute insight. Unfortunately, we did not take into consideration the Bayesian technique for meta-analysis in our research. In this case, we employed the classic frequentist strategy because we did not expect to see such a significant degree of heterogeneity. We would like to express our gratitude to the reviewer for drawing our attention to this innovative technique. When the study sizes are heterogeneous and the data are sparse, the Bayesian approach to meta-analysis appears to be a promising method of analysis. Considering the Bayesian technique in such a case is something we will look into in the future.

COMMENT. Minor issues

• I note some of the models for predicting hypertension use systolic blood pressure and diastolic blood pressure. Does this not appear ‘double-dipping’ to include a variable that may well be an outcome? Does this not require some comment?

RESPONSE: Thank you for noting this good point. The predictor systolic blood pressure and diastolic blood pressure are highly correlated with the outcome of hypertension. Please note that the models were used to predict incident (new-onset) hypertension. The people that the models were applied to did not have known hypertension at baseline. As would be expected, people with higher baseline blood pressure levels on the initial measurement were more likely to have sustained high blood pressure (or hypertension) long-term. While the predictor is highly correlated with the outcome, it is not synonymous with it.

COMMENT. • Page 13 line 330 – please be more specific than ‘basically’

RESPONSE: Thank you. We have changed the word now as suggested.

Please see page 13, line 316 in the revised manuscript.

COMMENT. • Page 15 line 388 – I belive it should be ‘models’

RESPONSE: Thank you. We have changed the word now as suggested.

Please see page 15, line 376 in the revised manuscript.

COMMENT. • Page 17 line 446-447 does not make sense

RESPONSE: Thank you. We have now removed the lines from the manuscript.

Please see page 17, lines 420-423 in the revised manuscript.

COMMENT. • Figure 1 – I believe the reasons for exclusion would be better ordered alphabetically or in descending number of records excluded.

RESPONSE: Thank you. We have now changed Figure 1. The reasons for exclusion are now presented in descending order on the number of records excluded.

Please see the revised figure 1.

6. PLOS authors have the option to publish the peer review history of their article (what does this mean?). If published, this will include your full peer review and any attached files.

Do you want your identity to be public for this peer review? For information about this choice, including consent withdrawal, please see our Privacy Policy.

Reviewer #1: Yes: John B. Kostis

Reviewer #2: No

---

## [Decision Letter · Decision Letter 1]

21 Mar 2022

Prediction of hypertension using traditional regression and machine learning models: A systematic review and meta-analysis

PONE-D-21-31564R1

Dear Dr. Turin,

We’re pleased to inform you that your manuscript has been judged scientifically suitable for publication and will be formally accepted for publication once it meets all outstanding technical requirements.

Kind regards,

Antonio Palazón-Bru, PhD

Academic Editor

PLOS ONE

Additional Editor Comments (optional):

Reviewers' comments:

Reviewer's Responses to Questions

**Comments to the Author**

1. If the authors have adequately addressed your comments raised in a previous round of review and you feel that this manuscript is now acceptable for publication, you may indicate that here to bypass the “Comments to the Author” section, enter your conflict of interest statement in the “Confidential to Editor” section, and submit your "Accept" recommendation.

Reviewer #1: All comments have been addressed

2. Is the manuscript technically sound, and do the data support the conclusions?

Reviewer #1: Yes

3. Has the statistical analysis been performed appropriately and rigorously? 

Reviewer #1: Yes

4. Have the authors made all data underlying the findings in their manuscript fully available?

Reviewer #1: Yes

5. Is the manuscript presented in an intelligible fashion and written in standard English?

Reviewer #1: Yes

6. Review Comments to the Author

Reviewer #1: In my opinion this manuscript is suitable for publication in PLOS ONE. The choice of the topic is timely and appropriate and the methodology used is correct in my opinion.

7. PLOS authors have the option to publish the peer review history of their article (what does this mean?). If published, this will include your full peer review and any attached files.

Reviewer #1: **Yes: **John B. Kostis

---

## [Editor Report · Acceptance letter]

28 Mar 2022

PONE-D-21-31564R1 

Prediction of hypertension using traditional regression and machine learning models: A systematic review and meta-analysis 

Dear Dr. Turin:

I'm pleased to inform you that your manuscript has been deemed suitable for publication in PLOS ONE. Congratulations! Your manuscript is now with our production department. 

Kind regards, 

on behalf of

Dr. Antonio Palazón-Bru 

Academic Editor

PLOS ONE